# The Statistical Fairness-Accuracy Frontier

**Alireza Fallah**[*]  **Michael I. Jordan**[†]  **Annie Ulichney**[‡]

## Abstract

Machine learning models must balance accuracy and fairness, but these goals often conflict, particularly when data come from multiple demographic groups with heterogeneous distributions. A useful tool for understanding this trade-off is the fairness-accuracy (FA) frontier, which characterizes the set of models that cannot be simultaneously improved in both fairness and accuracy. Prior analyses of the FA frontier provide a full characterization under the assumption of complete knowledge of population distributions—an unrealistic ideal. We study the FA frontier in the finite-sample regime, showing how sampling error and the heterogeneity in group distributions distort the empirical frontier from its population counterpart. In particular, we derive minimax-optimal estimators that depend on the designer's knowledge of the covariate distribution. For each estimator, we characterize how finite-sample effects asymmetrically impact each group's risk, and identify optimal sample allocation strategies. Our results transform the FA frontier from a theoretical construct into a practical tool for policymakers and practitioners who must often design algorithms with limited data.

## 1 Introduction

Across domains where predictive models guide consequential decisions—from lending and hiring to healthcare and education—society expects systems to be both *accurate* and *fair*. In supervised learning over a population composed of multiple subgroups with heterogeneous population distributions, these objectives are often in tension: models that minimize overall error can yield unequal performance across groups, while interventions that reduce disparities may incur accuracy losses. A central, practically relevant challenge arises when a *single predictive model* must serve multiple groups due to legal, logistical, or normative constraints. For instance, the Equal Credit Opportunity Act and Title VII of the Civil Rights Act in the United States discourage the use of group-specific models for credit and hiring decisions, respectively (see Raghavan et al. [46] for a discussion of guidelines on "disparate treatment").

In cases where group-specific models are disallowed or impractical, existing models are known to exhibit disparate performance across groups [5, 26, 19]; thus, understanding the best possible compromises between fairness and accuracy becomes critical. A principled way to approach this tension is through study of the *fairness–accuracy (FA) frontier*. The FA frontier consists of models where neither fairness nor accuracy can be improved without harming the other. This frontier allows decision-makers to visualize the explicit trade-offs and select a model that aligns with their fairness and performance objectives.

Recent work, notably by Liang et al. [37], characterizes the FA frontier under the assumption that group-wise distributions are known. In practice, however, we must base our decisions regarding the

---

[*]Department of Computer Science and Ken Kennedy Institute, Rice University

[†]Departments of Electrical Engineering and Computer Sciences and Statistics, University of California, Berkeley; Inria Paris

[‡]Department of Statistics, University of California, Berkeley

39th Conference on Neural Information Processing Systems (NeurIPS 2025) Workshop: Reliable ML from Unreliable Data.

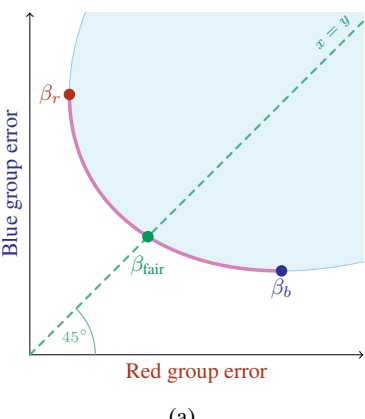
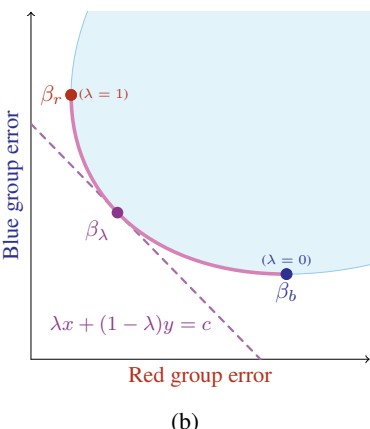

(a)                                                    (b)

Figure 1: (a) Population FA frontier: The blue shaded area represents the set of risk pairs that are achievable by some linear model, and $\beta_r$ and $\beta_b$ are the error-minimizing models for groups $r$ and $b$, respectively. $\beta_{\text{fair}}$ is the error-equalizing model. The FA frontier is the purple region along the boundary of the feasible set of error pairs. (b) For any $\lambda \in [0, 1]$, $\beta_\lambda$ is the first point of tangency between the line $\lambda x + (1 - \lambda)y = c$ and the FA frontier as we increase $c > 0$. This point moves along the frontier from $\beta_r$ to $\beta_b$ as $\lambda$ ranges from $1 \to 0$.

FA frontier on a finite number of data points. This work studies how the empirical FA frontier deviates from its population counterpart, characterizes the worst-case gap between them, and develops optimal estimators and sampling strategies.

We consider a two-group linear regression setting, with groups labeled "red" ($r$) and "blue" ($b$). Each individual's outcome in group $g \in \{r, b\}$ is modeled as the inner product of their covariate vector and a group-specific parameter $\beta_g$ plus zero-mean independent noise. The two groups may differ in their parameters ($\beta_r \neq \beta_b$) and in their covariate distributions. We measure the group-specific performance of a parameter $\beta$ by the expected squared loss, and the fairness by the absolute difference in risks– the *equalized loss* criterion [60, 32]. As presented by Liang et al. [37], Figure 1a depicts the FA frontier which traces a portion of the boundary of the set of all risk pairs (across the two groups) that are attainable by some linear model with endpoints at each group's optimum $\beta_g$.

For homoskedastic linear regression, we can parametrize the FA frontier using a parameter $\lambda \in [0, 1]$. For each $\lambda$, we define an objective function as a convex combination of group-specific risks, assigning weight $\lambda$ to the red group and $1 - \lambda$ to the blue group. The pair of risks corresponding to the minimizer of this objective, denoted by $\beta_\lambda$, lies on the FA frontier and represents the first point of tangency between the frontier and a line of the form $\lambda x + (1 - \lambda)y = c$ as $c > 0$ increases, as illustrated in Figure 1b. By varying $\lambda$ continuously from 0 to 1, we trace out the entire FA frontier. See Appendix A.0.1 for further discussion on the choice of $\lambda$.

Any estimator of a target model $\beta_\lambda$ will yield a risk pair that is strictly inside the achievable set. This raises a natural question: Given only $n_r$ and $n_b$ samples from groups $r$ and $b$, respectively, what estimator minimizes worst-case *excess risk*? To illustrate this, as shown in Figure 2a, consider lines of the form $\lambda x + (1 - \lambda)y = c$ and define the excess risk of an estimator as the amount by which $c$ must be increased so that it intersects an attainable risk pair, in the worst-case sense over a family of distributions.

First, we establish that the minimax-optimal estimator differs in the *known covariance* setting in which we know the covariance matrices of the covariate but the parameters $\beta_r$ and $\beta_b$ are unknown and the *unknown covariance* setting in which neither the covariance matrices nor the parameters are known. In the known covariance case, we derive matching upper and lower bounds for an estimator that preserves the structure of the oracle solution $\beta_\lambda$. We show that, under additional assumptions that control the tails of the covariate distributions, error increases in the norm of the group's covariance matrix and decreases with its sample size. This implies that higher-variance covariates require more samples to achieve the same accuracy and that increasing the sample size of only one group cannot reduce the overall error beyond a certain limit. In the unknown covariance case, we show that the ordinary least squares (OLS) estimator incurs risk consisting of *variance* and *bias* terms. The variance term arises from the estimation of the parameters $\beta_r$ and $\beta_b$, even when the covariance matrices are

known. The bias term, in contrast, represents the error incurred from the estimation of the covariance matrices and remains even when the group-wise parameters are known (see Proposition 2).

For subgaussian covariates satisfying the small-ball condition, we derive upper bounds for both the bias and variance terms. Our upper bound for the variance term naturally matches the bound in the known covariance case. Our upper bound for the bias term shows that it decreases in parameter heterogeneity $\|\beta_r - \beta_b\|$. We derive matching lower bounds (up to constants) to show that the OLS estimator is minimax-optimal.

Our bounds yield *optimal sampling rules* that identify how to allocate a fixed sampling budget in both cases. In the known-covariance case, the sampling allocation depends on the norm of the covariance matrix and weight $\lambda$[4]. In the unknown covariance case, when the variance term dominates (e.g., when $\|\beta_r - \beta_b\|$ is small), the same allocation rule as in the known covariance case remains near-optimal. When heterogeneity is large and the bias term dominates, however, the optimal design shifts toward *balancing* the two sample sizes, $n_r \approx n_b$ when the bias term dominates. The governing regime depends on $\|\beta_r - \beta_b\|$, i.e., in the heterogeneity across groups.

Finally, we demonstrate an asymmetry in the group-wise impact of finite-sample estimation, as depicted in Figure 2b. With known covariance, excess risk increases with ratio of the norms of their covariance matrices. With unknown covariance, the OLS estimator introduces a cross term that (i) can shift the risks in opposite directions and (ii) depends asymmetrically on $\lambda$. Notably, these disparities arise independently of sample size in both cases. Thus, the fairness–accuracy trade-off implied by the empirical estimator may differ substantially in practice from the trade-off defined at the population level.

**Related Work:**   Our work builds on Liang et al. [37], who formalize the FA Pareto frontier for assessing fairness-accuracy trade-offs under the assumption of full knowledge of the covariate and outcome distributions. Liu and Molinari [39] relax the full-knowledge assumption and introduce a consistent estimator of the FA frontier, deriving the asymptotic distribution of the estimator and using it to design statistical tests for properties such as the gap between a given algorithm and the fairest alternative. Auerbach et al. [4] further contribute a statistical test for determining whether a model achieving a given accuracy admits a Pareto improvement in fairness. We likewise address the practical challenges of analyzing the FA frontier from finite data, but take a fully non-asymptotic approach, providing finite-sample guarantees for estimators that target a social planner's fairness-accuracy preferences.

Our work also contributes to the literature on fairness in regression, a topic that has traditionally received less attention than fair classification [see, e.g., 21]. The most closely related work is that of Chzhen and Schreuder [12], who develop a minimax-optimal fair regression estimator for an explicit fairness constraint based on the Wasserstein distance between the loss distributions across groups. Beyond this, several other works have introduced various notions of fairness for the regression setting and have proposed corresponding optimal estimation procedures [8, 6, 25, 2, 13, 14, 45] and for estimating the minimal utility cost associated with achieving fairness in regression [62]. See Appendix A.1 for an extended discussion of related work.

## 2   Fairness-Accuracy Frontier Framework

We consider a population where each individual has covariates $X \in \mathcal{X} \subset \mathbb{R}^d$, outcome $Y \in \mathbb{R}$, and group label $g \in \mathcal{G} := \{r, b\}$ (e.g., red, blue). For each group $g$, we observe $n_g$ data points, $\mathcal{S}_g := \{(X_i^{(g)}, Y_i^{(g)})\}_{i=1}^{n_g}$ drawn i.i.d from $P(X, Y | G = g)$. Given a predictor $f \in \mathcal{H}$, its population and empirical risk on group $g$ under a loss function $\ell$ are given, respectively, by

$$\mathcal{R}_g(f) := \mathbb{E}_{(X,Y) \sim P(\cdot | G=g)}[\ell(f(X), Y)], \quad \widehat{\mathcal{R}}_g(f) := \frac{1}{n_g} \sum_{i=1}^{n_g} \ell\left(f\left(X_i^{(g)}\right), Y_i^{(g)}\right).$$

We measure fairness using the *risk disparity* $|\mathcal{R}_r(f) - \mathcal{R}_b(f)|$ as in [37]. For regression tasks with a continuous outcome variable $Y$—the primary focus of this paper—this notion coincides with the definition of *Equalized Loss* proposed by [60]. The set of achievable population risk pairs over the

---

[4]This result is a direct analogue of Neyman's classical allocation in stratified sampling [44]

class $\mathcal{H}$ is given by

$$\mathcal{E}(\mathcal{H}) := \{(\mathcal{R}_r(f), \mathcal{R}_b(f)) : f \in \mathcal{H}\}.$$

Following [37], we say that a point lies on the *fairness-accuracy (FA) Pareto frontier* if no other predictor simultaneously improves both groups' risks and risk disparity. See Appendix A.2 for extended discussion and definitions of the FA frontier.

In this work, we focus on the setting where there is a direct tradeoff between the accuracy across groups, referred to as the *group balanced* case in [37]. In the group-balanced setting, the *fair predictor* which minimizes the risk disparity falls between each group's optimal point on the frontier, denoted $f_r$ and $f_b$ for groups $r$ and $b$, respectively. In the next section, we show that the regression problem, under the assumption of equal noise variance, falls into the group-balanced category.

For the group-balanced setting, the FA frontier can be traversed by varying a parameter $\lambda \in [0, 1]$ which represents the designer's preference in trading off between fairness and accuracy for each group. In particular, given a weight $\lambda$, the decision-maker seeks a decision policy $f_\lambda$ that achieves the desired tradeoff in errors between groups. This corresponds to solving the weighted risk minimization:

$$f_\lambda = \arg\min_{f \in \mathcal{H}} \mathcal{R}_\lambda(f) \quad \text{with} \quad \mathcal{R}_\lambda(f) := \lambda \mathcal{R}_r(f) + (1 - \lambda)\mathcal{R}_b(f). \tag{1}$$

Here, $\lambda$ encodes the relative cost of errors affecting group $r$ to those affecting group $b$. The extreme cases $\lambda = 1$ and $\lambda = 0$ recover the group-optimal decision policies $f_r$ and $f_b$, respectively. In general, $f_\lambda$ is the first point in $\mathcal{E}(\mathcal{H})$ that intersects the line $\lambda \mathcal{R}_r + (1 - \lambda)\mathcal{R}_b = c$ as $c$ increases—that is, as the line shifts upward and to the right. See Figure 3a for an illustration. This formulation thus allows us to target a point on the FA frontier and trade off between fairness and accuracy.

## 2.1 Empirical Fairness-Accuracy Frontier

In practice, the true distribution $P$ is unknown, so $f_\lambda$ cannot be computed directly. Instead, we may learn an empirical predictor $\widehat{f}_\lambda \in \mathcal{H}$ from the data $\{\mathcal{S}_r, \mathcal{S}_b\}$. Its *excess risk*, given by $\mathcal{R}_\lambda(\widehat{f}_\lambda) - \mathcal{R}_\lambda(f_\lambda)$ is nonnegative, since $f_\lambda$ minimizes $\mathcal{R}_\lambda(\cdot)$. This implies that the empirical fairness-accuracy frontier contracts inward relative to the FA frontier $\mathcal{F}(\mathcal{H})$. That said, for any estimator $\widehat{f}_\lambda$, the corresponding excess risk quantifies how much the frontier is pushed inward in the direction orthogonal to the line $\lambda \mathcal{R}_r + (1 - \lambda)\mathcal{R}_b = $ constant, as illustrated in Figure 3b.

We study this contraction in a minimax sense. For a distribution class $\mathcal{P}$, we seek estimators minimizing the worst-case excess risk: $\inf_{\widehat{f}_\lambda \in \mathcal{H}} \sup_{P \in \mathcal{P}} \mathbb{E}\left[\mathcal{R}_\lambda(\widehat{f}_\lambda) - \mathcal{R}_\lambda(f_\lambda)\right]$ where the expectation is taken with respect to the draw of the dataset $\{\mathcal{S}_r, \mathcal{S}_b\}$. The remainder of this work analyzes this problem for linear regression, deriving lower bounds and near-optimal estimators and examining the group-wise risk implications.

## 3 Linear Model Class

To provide theoretical bounds on the empirical FA frontier, we focus on linear regression with homoskedastic noise.

**Definition 1** (Linear Model Class). *The linear model class $\mathcal{P}_{linear}(\sigma^2)$ consists of all distributions where, for each group $g \in \{r, b\}$,*

$$Y = X^\top \beta_g + \varepsilon_g, \quad \mathbb{E}[\varepsilon_g | X] = 0, \quad \mathbb{E}[\varepsilon_g^2 | X] = \sigma^2$$

*for some $\beta_g \in \mathbb{R}^d$. If we further assume that the distribution of the covariate $X$ given group identity $g$ is known and equal to $P_X^g$, then we denote this subclass of $\mathcal{P}_{linear}(\sigma^2)$ by $\mathcal{P}_{linear}(P_X^r, P_X^b, \sigma^2)$.*

A canonical example of a distribution in $\mathcal{P}_{\text{linear}}(\sigma^2)$ is the well-specified linear model with homoskedastic Gaussian noise $\varepsilon_g \sim \mathcal{N}(0, \sigma^2)$, independent of $X$. In what follows, we take $\mathcal{H}$ as the class of linear models, i.e., $\{f_\beta(x) = x^\top \beta : \beta \in \mathbb{R}^d\}$. In addition, we take the squared loss function $\ell(f_\beta(X), Y) = (\beta^\top X - Y)^2$, and also simplify the notation of the risks to $\mathcal{R}_g(f_\beta)$ and $\mathcal{R}_\lambda(f_\beta)$ to $\mathcal{R}_g(\beta)$ and $\mathcal{R}_\lambda(\beta)$, respectively. We write $\mathbb{E}_g$ and $\mathbb{P}_g$ to denote expectation and probability, respectively, under the conditional distribution $P(X, Y | G = g)$.

For group $g$, the population covariance is defined as $\Sigma_g := \mathbb{E}_g[XX^\top]$, and the cross-moment between $X$ and $Y$ is $\nu_g := \mathbb{E}_g[XY] = \Sigma_g \beta_g$. Their empirical quantities are defined, respectively, as

$$\widehat{\Sigma}_g := \frac{1}{n_g} \sum_{i=1}^{n_g} X_i^{(g)} X_i^{(g)\top}, \quad \widehat{\nu}_g := \frac{1}{n_g} \sum_{i=1}^{n_g} Y_i^{(g)} X_i^{(g)}.$$

The linear model falls under the group-balanced scenario described in Section 2. See Appendix A.3 for details.

**Assumptions** Assumptions 1 and 2 are maintained throughout our analysis and guarantee identifiability and consistency of the least-squares estimator. Assumptions 3 and 4 provide control over the tails of the sample covariance spectrum to further simplify our risk bounds.

**Assumption 1** (Invertible Covariance and Sample Covariance Matrix). *For each group $g \in \{r, b\}$, the population covariance matrix $\Sigma_g$ is invertible.*

**Assumption 2** (Invertible Sample Covariance Matrix). *For each group $g \in \{r, b\}$, the empirical covariance matrix $\widehat{\Sigma}_g$ is invertible almost surely.*

**Assumption 3** (Small-Ball Condition). *We say that the covariate of group $g$ satisfies the* small-ball condition *with parameters $C_g \geq 1$ and $\alpha_g \in (0, 1]$ if, for all nonzero $\theta \in \mathbb{R}^d$ and all $t > 0$,*

$$\mathbb{P}_g\left(|\theta^\top X| \leq t\,\|\theta\|_{\Sigma_g}\right) \leq (C_g t)^{\alpha_g}.$$

This condition, adopted from prior work (e.g., Koltchinskii and Mendelson [34], Mourtada [43]), provides lower-tail control of the sample covariance spectrum, ensuring that the empirical covariance does not degenerate in directions of low variance of the covariates. This assumption holds for multivariate Gaussian distributions with $\alpha_g = 1$, since in that case $\theta^\top X$ is Gaussian with variance $\|\theta\|_{\Sigma_g}^2$. Furthermore, [49] show that this condition holds with $\alpha_g = 1$ for covariates with independent coordinates and bounded densities.

**Assumption 4** (Subgaussian Covariates). *We say that the covariate of group $g$ satisfies the subgaussian assumption if there exists a constant $K_g \geq 1$ such that, for any $u \in \mathbb{R}^d$,*

$$\left\|X^\top u\right\|_{\psi_2} \leq K_g \left\|u\right\|_{\Sigma_g},$$

*where the $\psi_2$-norm is defined conditional on the group identity being $g$.*

This condition ensures that the covariates have light tails. It is satisfied, for instance, by multivariate Gaussian distributions or bounded distributions. Further insights about the complementary nature of Assumptions 3 and 4 are given in Appendix A.4.

## 3.1 Empirical Estimators

Let $\beta_\lambda$ denote the optimal predictor corresponding to the weighted loss $\mathcal{R}_\lambda(\beta)$,

$$\beta_\lambda := \underset{\beta \in \mathbb{R}^d}{\arg\min}\, \mathcal{R}_\lambda(\beta) \Sigma_\lambda^{-1} \nu_\lambda \tag{2}$$

with

$$\Sigma_\lambda := \lambda \Sigma_r + (1 - \lambda)\Sigma_b \quad \text{and} \quad \nu_\lambda := \lambda \nu_r + (1 - \lambda)\nu_b. \tag{3}$$

The next proposition allows us to relate the excess risk of any estimator $\beta$ to that of $\beta_\lambda$, as well as to its estimation error on both groups. For details, see Appendix C.1.1.

**Proposition 1.** *For a given $\beta \in \mathbb{R}^d$, $\lambda \in [0, 1]$, and $g \in \mathcal{G}$, the following identities hold:*

$$\mathcal{R}_\lambda(\beta) - \mathcal{R}_\lambda(\beta_\lambda) = \lambda \|\beta - \beta_\lambda\|_{\Sigma_r}^2 + (1 - \lambda)\|\beta - \beta_\lambda\|_{\Sigma_b}^2, \tag{4a}$$

$$\mathcal{R}_g(\beta) - \mathcal{R}_g(\beta_\lambda) = \|\beta - \beta_\lambda\|_{\Sigma_g}^2 + 2(\beta - \beta_\lambda)^\top \Sigma_g(\beta_\lambda - \beta_g). \tag{4b}$$

Recall that our goal is to minimize the worst case expected excess risk, i.e.,

$$\inf_\beta \sup_{P \in \mathcal{P}} \mathbb{E}_{\mathcal{S}_r, \mathcal{S}_b}\left[\mathcal{R}_\lambda(\beta) - \mathcal{R}_\lambda(\beta_\lambda)\right] = \inf_\beta \sup_{P \in \mathcal{P}} \mathbb{E}_{\mathcal{S}_r, \mathcal{S}_b}\left[\lambda\|\beta - \beta_\lambda\|_{\Sigma_r}^2 + (1 - \lambda)\|\beta - \beta_\lambda\|_{\Sigma_b}^2\right], \tag{5}$$

where $\beta$ is a function of the datasets $\{\mathcal{S}_r, \mathcal{S}_b\}$.[5] For $\lambda \in \{0, 1\}$, this reduces to the single-group setting. In that case, [43] shows that the OLS estimator $\widehat{\beta}_g$ is minimax optimal for $\mathcal{P}_{\text{linear}}(\sigma^2)$ and that knowledge of the covariate distribution does not affect this optimality. For $\lambda \in (0, 1)$, we consider the empirical analog of the $\lambda$-weighted risk: $\widehat{\mathcal{R}}_\lambda(\beta) := \lambda\widehat{\mathcal{R}}_r(\beta) + (1 - \lambda)\widehat{\mathcal{R}}_b(\beta)$. whose minimizer under Assumption 2 is

$$\widehat{\beta}_\lambda := \widehat{\Sigma}_\lambda^{-1}\widehat{\nu}_\lambda \quad \text{with} \quad \widehat{\Sigma}_\lambda = \lambda\widehat{\Sigma}_r + (1 - \lambda)\widehat{\Sigma}_b \quad \text{and} \quad \widehat{\nu}_\lambda := \lambda\widehat{\nu}_r + (1 - \lambda)\widehat{\nu}_b. \tag{6}$$

As we will see next, this estimator is nearly minimax optimal over $\mathcal{P}_{\text{linear}}(\sigma^2)$. However, unlike in the one-group case, under additional knowledge of the covariate distribution—or even just the covariance matrices $\Sigma_r$ and $\Sigma_b$—a different estimator becomes minimax optimal. This marks a sharp departure from the classical linear regression setting, in which the OLS estimator is optimal regardless of knowledge of the covariate distribution.

## 4  Estimation with Known Covariances

We first consider the setting in which the population covariance matrices $\Sigma_r$ and $\Sigma_b$ are known. Given samples from both groups, we propose the following estimator:

$$\tilde{\beta}_\lambda = (\Sigma_\lambda)^{-1}(\lambda\Sigma_r\widehat{\beta}_r + (1 - \lambda)\Sigma_b\widehat{\beta}_b), \tag{7}$$

with $\Sigma_\lambda$ as defined in (3) and $\widehat{\beta}_g$ defined as the standard, one-group OLS estimator. This estimator maintains the structure of the optimal predictor $\beta_\lambda$ but replaces the true cross moment $\nu_g = \Sigma_g\beta_g$ with the empirical quantity $\Sigma_g\widehat{\beta}_g$ for each group $g$. We first establish that this estimator is minimax optimal when the covariance matrices are known, and then characterize its group-wise estimation error. Finally, we discuss the implications of this result for algorithm design and optimal sampling strategies.

### 4.1  The Optimal Estimator

As Proposition 1 shows, the excess risk of an estimator can be decomposed into the sum of its distances from the optimal predictor $\beta_\lambda$ under the Mahalanobis norms induced by $\Sigma_r$ and $\Sigma_b$. The following result shows that, for each group $g$, the worst-case distance $\|\beta - \beta_\lambda\|_g^2$ is minimized by $\tilde{\beta}_\lambda$, thereby implying the minimax optimality of this estimator.

**Theorem 1.** *Suppose Assumptions 1 and 2 hold. Then, for any group $g \in \{b, r\}$, we have*

$$\inf_\beta \sup_{P \in \mathcal{P}_{linear}(P_X^r, P_X^b, \sigma^2)} \mathbb{E}\left[\|\beta - \beta_\lambda\|_{\Sigma_g}^2\right] \tag{8}$$

$$= \lambda^2\frac{\sigma^2}{n_r}\mathbb{E}\left[\text{Tr}\left(\Sigma_g\Sigma_\lambda^{-1}\Sigma_r\widehat{\Sigma}_r^{-1}\Sigma_r\Sigma_\lambda^{-1}\right)\right] + (1 - \lambda)^2\frac{\sigma^2}{n_b}\mathbb{E}\left[\text{Tr}\left(\Sigma_g\Sigma_\lambda^{-1}\Sigma_b\widehat{\Sigma}_b^{-1}\Sigma_b\Sigma_\lambda^{-1}\right)\right],$$

*and the infimum is achieved by setting $\beta$ equal to $\tilde{\beta}_\lambda$, given by (7).*

The proof is provided in Appendix C.2.1 and involves two steps: first, characterizing the error rate of the estimator $\tilde{\beta}_\lambda$ for any distribution in $\mathcal{P}_{\text{linear}}(P_X^r, P_X^b, \sigma^2)$; and second, showing that no other estimator can achieve a better rate in the worst-case sense via the Bayes estimator. Notice that we have only assumed knowledge of $\Sigma_g$ for each $g \in \{r, b\}$. This means that any additional knowledge of the distribution of $X$ for each group yields no estimation improvements. As stated in the discussion before Theorem 1, this result, together with Proposition 1, implies the following corollary on the minimax optimality of $\tilde{\beta}_\lambda$ with respect to the excess risk of $\mathcal{R}_\lambda(\cdot)$.

**Corollary 1.** *Suppose Assumptions 1 and 2 hold. Then, $\tilde{\beta}_\lambda$ is the minimizer of the worst-case excess risk (5) when $\mathcal{P}$ is set as $\mathcal{P}_{linear}(P_X^r, P_X^b, \sigma^2)$.*

### 4.2  Bounding the Excess Risk

In this subsection, to illustrate the implications of Theorem 1, we further simplify the error bound under additional assumptions on the distribution. First, note that when the covariance matrices are

---

[5] Moving forward, we omit the dependence of the expectation on the datasets when it is clear from the context.

known, it is without loss of generality to assume spherical covariances, since we can apply the transformation
$$\tilde{X}^{(g)} := \Sigma_g^{-1/2} X^{(g)}, \quad \tilde{\beta}_g := \Sigma_g^{-1/2} \beta_g,$$
which yields a spherical covariance matrix for the covariate vectors. We thus assume that each group has a spherical covariance structure, i.e., $\Sigma_g = \rho_g^2 I_d$ for known $\rho_g > 0$.

**Corollary 2.** *Suppose that for each group $g \in \mathcal{G}$, the covariance matrix satisfies $\Sigma_g = \rho_g^2 I_d$ for known $\rho_g > 0$. Furthermore, suppose Assumption 2 holds and the small-ball condition (Assumption 3) also holds with constants $(C_g, \alpha_g)$. Also suppose that $n_g \geq 6d/\alpha_g$ and that $d \geq 2$. Then we have*

$$\mathbb{E}\left[\left\|\tilde{\beta}_\lambda - \beta_\lambda\right\|_{\Sigma_g}^2\right] \leq \frac{2\sigma^2 d \rho_g^2}{(\lambda \rho_r^2 + (1-\lambda)\rho_b^2)^2}\left(\lambda^2 C_r' \frac{\rho_r^2}{n_r} + (1-\lambda)^2 C_b' \frac{\rho_b^2}{n_b}\right), \tag{9}$$

*where $C_g' = 3C_g^4 \exp(1 + 9/\alpha_g)$.*

See Appendix C.2.2 for the proof as well as Appendix A.5 for discussion of cases where the constant $C_g'$ can be further sharpened. The following corollary combines the differences in matrix norms to explicitly characterize the excess risk bound for the optimal estimator $\tilde{\beta}_\lambda$.

**Corollary 3.** *Under the premise of Corollary 2, we have*

$$\mathbb{E}\left[\mathcal{R}_\lambda(\tilde{\beta}_\lambda) - \mathcal{R}_\lambda(\beta_\lambda)\right] \leq \frac{2\sigma^2 d}{\lambda \rho_r^2 + (1-\lambda)\rho_b^2}\left(\lambda^2 C_r' \frac{\rho_r^2}{n_r} + (1-\lambda)^2 C_b' \frac{\rho_b^2}{n_b}\right). \tag{10}$$

We conclude this section with a few remarks on the insights we draw from these results.

**Optimal allocation of the sampling budget:** These results reveal the optimal allocation of a fixed sampling budget across the two groups. Corollary 2, along with a simple Cauchy–Schwarz inequality, suggests setting $\frac{n_r}{n_b} = \frac{\lambda \rho_r}{(1-\lambda)\rho_b}$ which intuitively increases with the group's weight $\lambda$ and scale $\rho_g$.

**Per-group estimation error:** Noting that $\tilde{\beta}_\lambda$ is an unbiased estimator of $\beta_\lambda$, by Proposition 1, we conclude that Theorem 1 and Corollary 2 quantify the per-group estimation error caused by using the empirical estimator instead of the true parameter. Since these terms differ only in the $\rho_g$ terms, we see that the asymmetry across groups is driven by differences in $\Sigma_g$ rather than $n_g$; When one group's sample size is low, it impacts the accuracy of the estimator $\tilde{\beta}_\lambda$, which, in turn, affects the error for both groups. An important implication of this discrepancy is that finite-sample estimation can distort the fairness-accuracy trade-off encoded by $\lambda$, shifting it away from the planner's intended balance.

# 5 Estimation with Unknown Covariance

We now turn to the setting where the covariances $\Sigma_r, \Sigma_b$ are unknown and must be estimated from data. Here, our estimator is the convex-combination OLS estimator:
$$\widehat{\beta}_\lambda = \widehat{\Sigma}_\lambda^{-1} \widehat{\nu}_\lambda = (\lambda \widehat{\Sigma}_r + (1-\lambda)\widehat{\Sigma}_b)^{-1}(\lambda \widehat{\nu}_r + (1-\lambda)\widehat{\nu}_b).$$
We first decompose its excess risk into bias and variance terms, then establish upper and lower bounds under subgaussian and small-ball conditions, showing near-optimality up to constants of the estimator under these assumptions.

## 5.1 The Bias-Variance Decomposition

Recall from Proposition 1 that the excess risk of $\widehat{\beta}_\lambda$ can be expressed as the weighted sum of the two distances $\left\|\widehat{\beta}_\lambda - \beta_\lambda\right\|_{\Sigma_r}^2$ and $\left\|\widehat{\beta}_\lambda - \beta_\lambda\right\|_{\Sigma_b}^2$. Thus, as in the previous section, we begin by characterizing these two distances.

**Proposition 2.** *Suppose Assumptions 1 and 2 hold. Then, for any group $g \in \{r, b\}$, we have:*

$$\mathbb{E}\left[\left\|\widehat{\beta}_\lambda - \beta_\lambda\right\|_{\Sigma_g}^2\right] \leq \mathcal{V}_g(\lambda) + \mathcal{B}_g(\lambda), \tag{11}$$

*where the variance and bias terms are given, respectively, by:*

$$\mathcal{V}_g(\lambda) := \lambda^2 \frac{\sigma^2}{n_r} \mathbb{E} \operatorname{Tr}\left(\widehat{\Sigma}_\lambda^{-1} \Sigma_g \widehat{\Sigma}_\lambda^{-1} \widehat{\Sigma}_r\right) + (1-\lambda)^2 \frac{\sigma^2}{n_b} \mathbb{E} \operatorname{Tr}\left(\widehat{\Sigma}_\lambda^{-1} \Sigma_g \widehat{\Sigma}_\lambda^{-1} \widehat{\Sigma}_b\right), \tag{12}$$

$$\mathcal{B}_g(\lambda) := \mathbb{E}\left[\left\|\Sigma_g^{1/2}\left[\left(I_d + \frac{1-\lambda}{\lambda}\widehat{\Sigma}_r^{-1}\widehat{\Sigma}_b\right)^{-1} - \left(I_d + \frac{1-\lambda}{\lambda}\Sigma_r^{-1}\Sigma_b\right)^{-1}\right]\right\|^2\right] \|\beta_r - \beta_b\|^2. \tag{13}$$

See Appendix C.3.1 for the proof. The variance term closely resembles the error rate in the known-covariance case: it captures the irreducible sampling error within each group, as it is scaled by the squared fairness weights and the inverse sample sizes of each group. The bias term, however, represents a key departure from the known-covariance setting. It arises from heterogeneity in the true underlying coefficients $\beta_r$ and $\beta_b$, it depends explicitly on $\beta_r - \beta_b$, and it vanishes when $\beta_r = \beta_b$. Even if $\beta_r$ and $\beta_b$ are known exactly, the target parameter $\beta_\lambda$ cannot be recovered without knowledge of $\Sigma_r, \Sigma_b$. The bias term quantifies the additional error introduced by replacing the true covariance structure with empirical estimates, which explains why it vanishes in the known-covariance case.

## 5.2 Upper and Lower Bounds for the Bias and Variance Terms

In this section, we investigate the optimality of the estimator $\widehat{\beta}_\lambda$ under additional assumptions. For our upper bounds, we consider subgaussian covariates (Assumption 4) that satisfy the small-ball condition (Assumption 3). We further assume that, for any group $g \in \{r, b\}$,

$$\frac{1}{2}\rho_g^2 I_d \preceq \Sigma_g \preceq \frac{3}{2}\rho_g^2 I_d, \quad |\beta_g| \leq B. \tag{14}$$

In the first condition, constants $1/2$ and $3/2$ are chosen for simplicity, and the appendix gives a more general upper bound. We impose the second condition since Proposition 2 (and later our lower bound) shows that the bias term grows with $\|\beta_r - \beta_b\|$ and would diverge if the groups' predictors were unbounded. Accordingly, we consider the subclass of $\mathcal{P}_{\text{linear}}(\sigma^2)$ consisting of Gaussian covariates satisfying conditions (14), and we denote the class by $\mathcal{P}_{\text{Gauss}}(\sigma^2, \rho_r^2, \rho_b^2, B)$.

For the upper bound, we bound the quantities $\mathcal{V}_g(\lambda)$ and $\mathcal{B}_g(\lambda)$ from Proposition 2. For the lower bound, we decompose the worst-case error into bias and variance components by considering two complementary scenarios. Specifically, we lower bound

$$\sup_{P \in \mathcal{P}_{\text{Gauss}}(\sigma^2, \rho_r^2, \rho_b^2, B)} \mathbb{E}\left[\|\beta - \beta_\lambda\|_{\Sigma_g}^2\right] \tag{15}$$

by the maximum of two restricted subproblems: (1) the case where the covariance matrices $\Sigma_r$ and $\Sigma_b$ are known but the group predictors $\beta_r$ and $\beta_b$ are unknown (corresponding to the variance term), and (2) the case where the group predictors are known but the covariance matrices are unknown (corresponding to the bias term).

**Theorem 2.** *Suppose Assumptions 1–4 hold, the covariance matrices satisfy* (14)*, and, for $g \in \mathcal{G}$, and $n_g \geq \max\{48/\alpha_g, K_g^4 d\}$. Then, we have:*

$$\mathcal{V}_g(\lambda) \lesssim \frac{\rho_g^2 \sigma^2 d}{(\lambda\rho_r^2/C_r' + (1-\lambda)\rho_b^2/C_b')^2} \left(\frac{\lambda^2 \rho_r^2}{n_r} + \frac{(1-\lambda)^2 \rho_b^2}{n_b}\right), \tag{16}$$

*where $\mathcal{V}_g(\lambda)$ is the variance term, as defined in Proposition 2, and $C_g'$ is given in Corollary 2. Moreover, assuming $n_g \geq \sigma^2/(B\rho_g^2)$ for $g \in \mathcal{G}$, we have*

$$\inf_{\beta} \sup_{\substack{P \in \mathcal{P}_{Gauss}(\sigma^2, \rho_r^2, \rho_b^2, B) \\ \Sigma_r, \Sigma_b \text{ are known.}}} \mathbb{E}\left[\|\beta - \beta_\lambda\|_{\Sigma_g}^2\right] \gtrsim \frac{\rho_g^2 \sigma^2 d}{(\lambda\rho_r^2 + (1-\lambda)\rho_b^2)^2} \left(\frac{\lambda^2 \rho_r^2}{n_r} + \frac{(1-\lambda)^2 \rho_b^2}{n_b}\right), \tag{17}$$

*where the infimum is taken over any estimator $\beta$ as a function of the datasets $(\mathcal{S}_r, \mathcal{S}_b)$.*

See Appendix C.3.2 for the proof. The upper and lower bounds together show that, up to constant factors, the variance term indeed captures the error arising from not knowing the true predictors $\beta_r$ and $\beta_b$, even in the known-covariance case. Moreover, the estimator $\widehat{\beta}_\lambda$ is minimax-optimal, again up to constant factors. Note that, while the lower bound is for the known-covariance case, the result in Theorem 1 from the previous section do not apply here since we have assumed bounded parameters. We next derive upper and lower bounds for the bias term.

**Theorem 3.** *Suppose Assumptions 1–4 hold, the covariance matrices satisfy* (14)*, and, for $g \in \mathcal{G}$, $n_g \geq 48/\alpha_g$. Then, we have:*

$$\mathcal{B}_g(\lambda) \lesssim \frac{\lambda^2 (1-\lambda)^2 \rho_g^2 \rho_r^4 \rho_b^4 d}{(\lambda\rho_r^2/C_r' + (1-\lambda)\rho_b^2/C_b')^2 (\lambda\rho_r^2 + (1-\lambda)\rho_b^2)^2} \left(\frac{K_r^4}{n_r} + \frac{K_b^4}{n_b}\right) \|\beta_r - \beta_b\|^2, \tag{18}$$

*where $\mathcal{B}_g(\lambda)$ is the bias term, as defined in Proposition 2, and $C_g'$ is given in Corollary 2. Moreover, if $n_g \geq 16d^2$ for $g \in \mathcal{G}$, we have*

$$\inf_{\beta} \sup_{\substack{P \in \mathcal{P}_{Gauss}(\sigma^2, \rho_r^2, \rho_b^2, B) \\ \beta_r, \beta_b \text{ are known.}}} \mathbb{E}\left[\|\beta - \beta_\lambda\|_{\Sigma_g}^2\right] \gtrsim \frac{\lambda^2 (1-\lambda)^2 \rho_g^2 \rho_r^4 \rho_b^4 d}{(\lambda\rho_r^2 + (1-\lambda)\rho_b^2)^4} \left(\frac{1}{n_r} + \frac{1}{n_b}\right) \|\beta_r - \beta_b\|^2, \tag{19}$$

*where the infimum is taken over any estimator $\beta$ as a function of the datasets $(\mathcal{S}_r, \mathcal{S}_b)$.*

See Appendix C.3.3 for the proof. Recall that the constants $K_r$ and $K_b$, as defined in Assumption 4, are invariant to scaling and hence are independent of $\rho_r$ and $\rho_b$. As a result, the upper and lower bounds above match up to constant factors, which again shows that the bias term truly captures the error arising from not knowing the covariance matrices, even when the group predictors $\beta_r$ and $\beta_b$ are known. Moreover, Theorem 3, together with the result of Theorem 2, highlights that the OLS estimator $\widehat{\beta}_\lambda$ achieves the minimax excess risk, as given in (5) with $\mathcal{P} = \mathcal{P}_{\text{Gauss}}(\sigma^2, \rho_r^2, \rho_b^2, B)$, up to constant factors.

**Optimal allocation of the sampling budget:** Similar to the known covariance setting, under a fixed total sampling budget, these results imply that we should choose $n_r/n_b = (\lambda\rho_r)/((1-\lambda)\rho_b)$ to minimize the variance term. In contrast, Theorem 3 shows that minimizing the bias term requires a balanced design, namely, $n_r = n_b$. The optimal sampling allocation strategy is more nuanced here and depends on whether the variance term of bias term dominates. This, in turn, depends on the heterogeneity in group distributions: the more the two groups differ, the larger the bias term becomes, and the more we would prefer $n_r$ and $n_b$ to be closer.

### 5.3 Per-Group Estimation Errors

In the known covariance case (see Section 4), the cross term in the group-wise excess risk given by Proposition 1 is mean-zero, so our bounds on $\|\widehat{\beta}_\lambda - \beta_\lambda\|_{\Sigma_g}^2$ translate directly to the per-group (expected) estimation error. However, this property does not hold for unknown covariance, as $\widehat{\beta}_\lambda$ is not an unbiased estimate of $\beta_\lambda$. The following result bounds the cross term in Proposition 1:

**Proposition 3.** *Suppose Assumptions 1–4 hold, the covariance matrices satisfy* (14)*, and, for $g \in \mathcal{G}$, $n_g \geq 48/\alpha_g$. Then, we have:*

$$
\left| \mathbb{E}\left[ (\widehat{\beta}_\lambda - \beta_\lambda)^\top \Sigma_r (\beta_\lambda - \beta_r) \right] \right| \lesssim
$$

$$
\frac{\lambda(1-\lambda)^2 d\rho_r^4\rho_b^4 \, \|\beta_r - \beta_b\|^2}{(\lambda\rho_r^2 + (1-\lambda)\rho_b^2)^3 \, (\lambda\rho_r^2/C_r' + (1-\lambda)\rho_b^2/C_b')} \left( \frac{K_r^2}{\sqrt{n_r}} + \frac{K_b^2}{\sqrt{n_b}} \right) \left( \lambda\frac{K_r^2\rho_r^2}{\sqrt{n_r}} + (1-\lambda)\frac{K_b^2\rho_b^2}{\sqrt{n_b}} \right),
$$

*where $C_g'$ is given in Corollary 2.*

The proof is provided in Appendix C.3.4. A similar result can also be stated for group $b$, with the only difference being that $\lambda(1 - \lambda)^2$ in the bound is replaced by $\lambda^2(1 - \lambda)$. Proposition 3, Theorem 2 and Theorem 3, show that, much like in the known-covariance setting, there is an inherent asymmetry in how the empirical estimator affects the risks of the two groups. The asymmetry revealed in Proposition 1 mirrors that of the known-covariance setting in that it is driven only by $\rho_g$.

The cross term, $(\widehat{\beta}_\lambda - \beta_\lambda)^\top \Sigma_g (\beta_\lambda - \beta_g)$, captures a new phenomenon arising from the bias term. First, note that the bound in Proposition 3 controls its absolute value, but the term itself may be positive or negative. Moreover, it is straightforward to verify that if we sum this term across the two groups with weights $\lambda$ and $1 - \lambda$, the result is zero. Thus, its contribution to one group's risk is always offset by the other's, ensuring that the two groups experience it with opposite signs. Even if the absolute-value bound were symmetric, the term would still be a source of disparity between the two groups' risks. However, the bound is in fact not symmetric. As stated after the proposition, it changes from $\lambda(1 - \lambda)^2$ for group $r$ to $\lambda^2(1 - \lambda)$ for group $b$. This bias effect is larger for group $r$ when $\lambda$ is close to 0 and for group $b$ when $\lambda$ is close to 1, meaning that the second term increases in absolute value for one group when the other group is prioritized in the overall loss function.

The fairness-accuracy trade-off implied by the choice of $\lambda$ in the population objective is not necessarily the trade-off realized in finite samples. The systematic differences in per-group risk may shift the balance in fairness and accuracy away from the intended allocation, however, as in the known-covariance setting, all bounds remain similar in $n_r$ and $n_b$ across both groups.

## Acknowledgments and Disclosure of Funding

We are grateful to Annie Liang for insightful comments and discussion. Annie Ulichney's work is supported by the National Science Foundation Graduate Research Fellowship Program under Grant

No. DGE 2146752. We also acknowledge funding from the European Union (ERC-2022-SYG-OCEAN-101071601). Views and opinions expressed are however those of the author(s) only and do not necessarily reflect those of the National Science Foundation, the European Union or the European Research Council Executive Agency.

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

## A   Additional Discussion

### A.0.1   Choosing the Parameter $\lambda$

The choice of $\lambda$ is a central design decision shaped by the model's legal and ethical context, reflecting the social planner's preferences over group errors. To illustrate this, suppose 90% of the population belongs to the red group. A natural choice would then be to set $\lambda = 0.9$. However, if we are training a predictive model for patients or for hiring across different demographics, we might want to ensure that the smaller group is not overshadowed. In that case, we could increase its weight and even assign equal importance to both groups, i.e., set $\lambda = 0.5$. This approach, often referred to as "*weighting by inverse class frequency*," is common in the fairness literature [18, 52]. There are also smoothed variants that interpolate between these extremes, offering a context-dependent balance between accuracy and fairness [42]. In summary, the choice of $\lambda$ ultimately rests with the designer, and the role of the FA frontier is to ensure that, whatever choice is made, the resulting algorithm is optimal for that setting.

### A.1   Additional Related Work

A growing body of work bridges perspectives on fairness-accuracy trade-offs in computer science and statistics with analyses of social welfare [30, 36, 27, 48]. Our results on the FA frontier can also be interpreted in this context if we view each individual's utility as the risk associated with their group. More broadly, in the algorithmic fairness literature, it is widely recognized that no single fairness criterion can capture the diverse preferences that decision-makers may have over these trade-offs [10, 53, 33]. Economists make this point in a similar way by documenting the heterogeneity in how individuals weigh equity against efficiency [9, 40]. Our framework builds on this connection by adopting a flexible, weight-based formulation that accommodates this kind of heterogeneity [50, 29, 31].

Finally, much attention has been devoted to the fundamental question of when fairness and accuracy are in conflict and when they are aligned [16, 57, 7, 37, 3, 22, 15]. In settings where these objectives are at odds, a variety of methods have been developed to identify *Pareto improvements*, where either fairness or accuracy can be improved while keeping the other one unchanged. One line of work designs algorithms for computing fairer alternatives to a given model for specific applications [24, 17, 38, 56, 11, 20]. A closely related line of work draws on the legal notion of a *Less Discriminatory Alternative*, or a fairer model that maintains the original model's accuracy [28, 35, 51]. Others take the reverse approach, focusing on accuracy improvements subject to fairness guarantees [1, 55], or, more generally, methods for balancing multiple competing objectives in learning [47, 41].

## A.2 Details on Defining the FA Frontier

Here we recall the definition of the *fairness-accuracy (FA) Pareto frontier* from [37]. To do so, we first revisit the definition of *FA dominance*.

**Definition 2** (Fairness-Accuracy (FA) Dominance). *We say that a function $f' \in \mathcal{H}$ FA-dominates a function $f \in \mathcal{H}$, denoted by $f' \succ f$, if $\mathcal{R}_r(f') \leq \mathcal{R}_r(f)$, $\mathcal{R}_b(f') \leq \mathcal{R}_b(f)$, and $|\mathcal{R}_r(f') - \mathcal{R}_b(f')| \leq |\mathcal{R}_r(f) - \mathcal{R}_b(f)|$, with at least one of these three inequalities being strict.*

In other words, a function $f'$ FA-dominates a function $f$ if it achieves no higher risk on either group and no greater risk disparity between groups, with strict improvement for at least one. The FA frontier is then defined as the subset of achievable risk pairs that are not FA-dominated by any other point.

**Definition 3** (Fairness-Accuracy (FA) Frontier). *The FA frontier, denoted by $\mathcal{F}(\mathcal{H})$, is defined as:*

$$\mathcal{F}(\mathcal{H}) \coloneqq \{(\mathcal{R}_r(f), \mathcal{R}_b(f)) \in \mathcal{E}(\mathcal{H}) : \nexists f' \in \mathcal{H} : f' \succ f\}.$$

## A.3 Formalizing Group-Balanced Setting in Linear Regression

**Lemma 1** (Group-Balanced Structure). *The linear model described above exhibits a* group-balanced *structure. That is, each group's risk-minimizing predictor achieves (weakly) lower prediction error on its own group than on the other:*

$$\mathcal{R}_r(\beta_r) \leq \mathcal{R}_b(\beta_r), \quad \mathcal{R}_r(\beta_b) \geq \mathcal{R}_b(\beta_b).$$

### A.3.1 Proof of Lemma 1

For any group $g \in \{r, b\}$, the risk of the group-optimal predictor $\beta_g$ is:

$$\mathcal{R}_g(\beta_g) = \mathbb{E}_g \left[ (X^\top \beta_g - Y)^2 \right] = \mathbb{E}_g[\varepsilon_g^2] = \sigma^2.$$

For the other group $g' \neq g$, the same predictor incurs risk:

$$\mathcal{R}_{g'}(\beta_g) = \mathbb{E}_{g'} \left[ (X^\top \beta_g - Y)^2 \right] = \mathbb{E}_{g'} \left[ \left( X^\top (\beta_g - \beta_{g'}) \right)^2 \right] + \sigma^2.$$

The latter is greater or equal, and strictly greater when $\beta_g \neq \beta_{g'}$ and $\Sigma_{g'}$ is invertible.

## A.4 The Role of Concentration and Anticoncentration Assumptions

Assumptions 3 and 4 are complementary and provide control over the spectrum of the sample covariance matrix. The subgaussian condition bounds the upper tail, limiting the growth of the largest eigenvalue and thereby the variance of our estimators. The small-ball condition controls the lower tail, preventing the collapse of the smallest eigenvalue such that inverse covariance terms are bounded. As discussed above, the two assumptions hold simultaneously for a broad class of distributions, including multivariate Gaussian distributions and covariates with independent subgaussian coordinates and bounded density.

## A.5 Sharpening Analysis of Minimum Eigenvalue of the Sample Covariance Matrix

The constant $C_g'$ arises from the bound on the minimum eigenvalue of the sample covariance, which is used here to bound the trace of $\widehat{\Sigma}_r^{-1}$ and $\widehat{\Sigma}_b^{-1}$, as given in Mourtada [43, Theorem 4]. However, as highlighted in Mourtada [43, Remark 6]—based on results from Wu and Verdú [58] and Edelman [23]—this constant can be significantly reduced for Gaussian distributions. In particular, if the covariate distributions are Gaussian, $C_g'$ can be replaced by a constant whose limit, when $d/n_g \to h \in (0, 1)$, is upper bounded by

$$\left( \frac{1}{h} \right)^{3h} \left( \frac{\sqrt{e}}{1 - h} \right)^{3(1-h)},$$

which, in turn, can be shown to be upper bounded by $(1 + \sqrt{e})^3$.

## B  Additional Figures

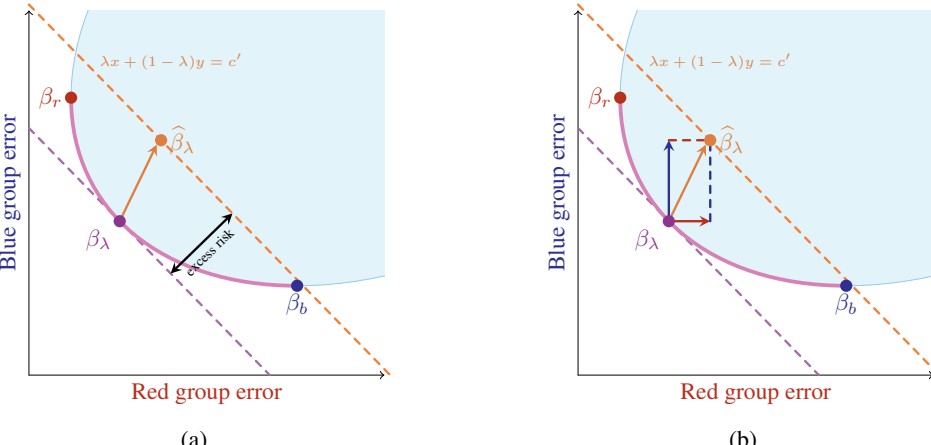

Figure 2: (a) Finite-sample estimation: The error pair corresponding to the empirical estimator $\widehat{\beta}_\lambda$ lies on the line $\lambda x + (1-\lambda)y = c'$, where $c - c'$ is the excess risk. (b) Asymmetric group-wise impact: The displacement from $\beta_\lambda$ to $\widehat{\beta}_\lambda$ decomposes into a vertical change in the blue-group error and a horizontal change in the red-group error; their unequal magnitudes show that the estimator affects the two groups asymmetrically.

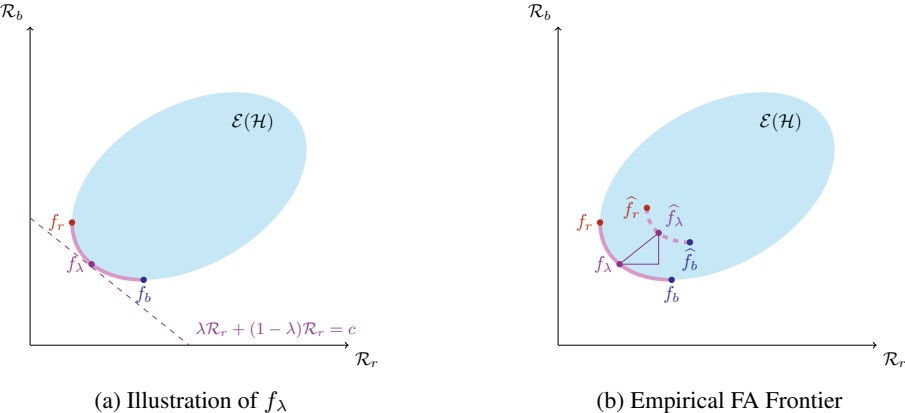

(a) Illustration of $f_\lambda$            (b) Empirical FA Frontier

Figure 3: Illustration of FA frontier, parametrized by $\lambda$, and its empirical version.

## C  Additional Results and Proofs

### C.0.1  Auxiliary Results

**Lemma 2** (Exact risk of the OLS convex combination estimator)**.** *Under Definition 1 and Assumptions 1, 2 the excess risk decomposes as:*

$$\mathbb{E}\left[\left\|\widehat{\beta}_\lambda - \beta_\lambda\right\|_{\Sigma_g}^2\right] = \lambda^2 \frac{1}{n_r^2}\mathbb{E}\left[\sum_{i=1}^{n_r}\sigma_r^2(X_i)\left\|\tilde{X}_i\right\|_{\tilde{\Sigma}_g^{-2}}^2\right]$$

$$+ (1-\lambda)^2\frac{1}{n_b^2}\mathbb{E}\left[\sum_{i=1}^{n_b}\sigma_b^2(X_i)\left\|\tilde{X}_i\right\|_{\tilde{\Sigma}_g^{-2}}^2\right]$$

$$+ \quad \mathbb{E}\left[\left\|\lambda\widehat{\Sigma}_r(\beta_r - \beta_\lambda) + (1-\lambda)\widehat{\Sigma}_b(\beta_b - \beta_\lambda)\right\|_{\tilde{\Sigma}_g^{-2}}^2\right].$$

**Proof of Lemma 2**

We begin by noting that

$$\widehat{\nu}_g := \frac{1}{n_g} \sum_{i=1}^{n_g} X_i Y_i = \widehat{\Sigma}_g \beta_g + \frac{1}{n_g} \sum_{i=1}^{n_g} \varepsilon_i X_i.$$

Substituting into $\widehat{\nu}_\lambda$, we obtain

$$\lambda \widehat{\nu}_r + (1 - \lambda) \widehat{\nu}_b = \lambda \widehat{\Sigma}_r \beta_r + (1 - \lambda) \widehat{\Sigma}_b \beta_b + \lambda \frac{1}{n_r} \sum_{i=1}^{n_r} \varepsilon_i X_i + (1 - \lambda) \frac{1}{n_b} \sum_{i=1}^{n_b} \varepsilon_i X_i.$$

Define the symmetric matrix

$$W := \widehat{\Sigma}_\lambda^{-1} \Sigma_g \widehat{\Sigma}_\lambda^{-1}$$

and let

$$Z_r := \lambda \frac{1}{n_r} \sum_{i=1}^{n_r} \varepsilon_i X_i, \quad Z_b := (1 - \lambda) \frac{1}{n_b} \sum_{i=1}^{n_b} \varepsilon_i X_i, \quad Z := Z_r + Z_b$$

$$B = \lambda \widehat{\Sigma}_r (\beta_r - \beta_\lambda) + (1 - \lambda) \widehat{\Sigma}_b (\beta_b - \beta_\lambda).$$

The error in estimating $\beta_\lambda$ thus decomposes as

$$\widehat{\beta}_\lambda - \beta_\lambda = \widehat{\Sigma}_\lambda^{-1} (Z + B).$$

so that

$$\mathbb{E} \left[ \left\| \widehat{\beta}_\lambda - \beta_\lambda \right\|_{\Sigma_g}^2 \right] = \mathbb{E} \left[ \| Z + B \|_W^2 \right] = \mathbb{E} \left[ \| Z \|_W^2 \right] + \mathbb{E} \left[ \| B \|_W^2 \right] + 2 \mathbb{E} \left[ \langle Z, B \rangle_W \right].$$

Since $Z$ is mean-zero conditional on $\{X_i\}$ and $B$ and $W$ are deterministic given $\{X_i\}$, we have

$$\mathbb{E} \left[ \mathbb{E} \left[ \langle Z, B \rangle_W \mid \{X_i\} \right] \right] = 0.$$

The cross-terms between $Z_r$ and $Z_b$ also vanish by independence and the fact that the noise is zero-mean:

$$\mathbb{E} \left[ \mathbb{E} \left[ \langle Z_r, Z_b \rangle_W \right] \mid \{X_i\} \right] = 0.$$

Therefore,

$$\mathbb{E} \left[ \| Z \|_W^2 \right] = \mathbb{E} \left[ \| Z_r \|_W^2 \right] + \mathbb{E} \left[ \| Z_b \|_W^2 \right]$$

$$= \lambda^2 \frac{1}{n_r^2} \mathbb{E} \left[ \sum_{i=1}^{n_r} \sigma_r^2(X_i) \| X_i \|_W^2 \right] + (1 - \lambda)^2 \frac{1}{n_b^2} \mathbb{E} \left[ \sum_{i=1}^{n_b} \sigma_b^2(X_i) \| X_i \|_W^2 \right]$$

Combining terms and observing that

$$\| X_i \|_W^2 = \left\| \tilde{X}_i \right\|_{\tilde{\Sigma}_g^{-2}}$$

yields the claim. ∎

**Lemma 3** ([43] Corollary 4 generalized to spherical covariance). *Suppose that $n_g \geq 48/\alpha_g$ and that $X^{(g)}$ satisfies Assumption 3 with parameters $C_g$ and $\alpha_g$. Also suppose that $\rho_g^2 I_d \preceq \Sigma_g$. Then,*

$$\mathbb{E} \left[ \lambda_{\min}(\widehat{\Sigma}_g)^{-4} \right] \leq 2 \rho_g^{-8} \cdot C_g'^4 \tag{20}$$

*where $C_g' := 3 C_g^4 \exp(1 + 9/\alpha_g)$.*

**Proof of Lemma 3**

Define the whitened covariate vectors and whitened sample covariance, respectively, as

$$\tilde{X}_i^{(g)} := \frac{1}{\rho_g} X_i^{(g)}, \quad \bar{\Sigma}_g := \frac{1}{n_g} \sum_{i=1}^{n_g} \tilde{X}_i^{(g)} \tilde{X}_i^{(g)\top}.$$

By [43] Corollary 4 taking $q = 4$, if $n_g \geq 48/\alpha_g$,

$$\mathbb{E}\left[\lambda_{\min}(\bar{\Sigma}_g)^{-4}\right] \leq 2 \cdot \left(3C_g^4 \exp(1 + 9/\alpha_g)\right)^4.$$

Hence,

$$\mathbb{E}\left[\lambda_{\min}(\widehat{\Sigma}_g)^{-4}\right] \leq 2\rho_g^{-8} \cdot \left(3C_g^4 \exp(1 + 9/\alpha_g)\right)^4. \blacksquare \tag{21}$$

**Lemma 4** (Subgaussian sample covariance moment bound). *Under Assumption 4, there exists a universal constant $C > 0$ such that:*

$$\mathbb{E}\left[\left\|\widehat{\Sigma}_g - \Sigma_g\right\|^4\right] \leq CK_g^8 \|\Sigma_g\|^4 \frac{d^2}{n_g^2}.$$

**Proof of Lemma 4**

By standard subgaussian matrix concentration (see, e.g., [54] Remark 4.7.3), for all $\delta \in (0,1)$, there exists an absolute constant $C > 0$ such that:

$$\mathbb{P}\left(\left\|\widehat{\Sigma}_g - \Sigma_g\right\| \geq CK_g^2\left(\sqrt{\frac{d+u}{n_g}} + \frac{d+u}{n_g}\right)\|\Sigma_g\|\right) \leq 2\exp(-u).$$

To invert this tail bound, we seek $f(s)$ such that $\mathbb{P}(\|\widehat{\Sigma}_g - \Sigma_g\| \geq s) \leq 2\exp(-f(s))$. In other words, using $2\max\{a,b\} \geq a + b$, we want

$$s \leq CK_g^2\left(\sqrt{\frac{d+u}{n_g}} + \frac{d+u}{n_g}\right)\|\Sigma_g\| \leq 2CK_g^2\max\left\{\sqrt{\frac{d+u}{n_g}}, \frac{d+u}{n_g}\right\} \tag{22}$$

We split this into two cases:

1. If $\sqrt{\frac{d+u}{n_g}} \geq \frac{d+u}{n_g}$, (22) is satisfied if $s \leq 2CK_g^2\sqrt{\frac{d+u}{n_g}}$ which occurs when $u \geq \frac{s^2 n_g}{4C^2 K_g^4 \|\Sigma_g\|^2} - d$.

2. If $\sqrt{\frac{d+u}{n_g}} < \frac{d+u}{n_g}$, (22) is satisfied if $s \leq 2CK_g^2\sqrt{\frac{d+u}{n_g}}$ which occurs when $u \geq \frac{sn_g}{2CK_g^2\|\Sigma_g\|} - d$.

Combining,

$$\mathbb{P}\left(\left\|\widehat{\Sigma}_g - \Sigma_g\right\| \geq s\right) \leq 2\exp\left(d - C'n_g \min\left\{\frac{s^2}{K_g^4\|\Sigma_g\|^2}, \frac{s}{K_g^2\|\Sigma_g\|}\right\}\right).$$

Define $s_1 := K_g^2\|\Sigma_g\|\sqrt{\frac{d}{C'n_g}}$ and $s_2 := K_g^2\|\Sigma_g\|\frac{d}{C'n_g}$, the critical values of $s$ that make the argument of the exponent positive for cases (1) and (2) defined above, respectively.

Next, we express the expectation as the integral of the tails, performing the change of variables $t = s^4$:

$$\mathbb{E}\left[\left\|\widehat{\Sigma}_g - \Sigma_g\right\|^4\right] = \int_0^\infty \mathbb{P}\left(\left\|\widehat{\Sigma}_g - \Sigma_g\right\|^4 \geq t\right) dt = 4\int_0^\infty \mathbb{P}\left(\left\|\widehat{\Sigma}_g - \Sigma_g\right\| \geq s\right) s^3 ds$$

Define the critical point $s_d := \min\{s_1, s_2\}$ and split the integral

$$\int_0^\infty \mathbb{P}\left(\left\|\widehat{\Sigma}_g - \Sigma_g\right\| \geq s\right) s^3 ds = \int_0^{s_d} \mathbb{P}\left(\left\|\widehat{\Sigma}_g - \Sigma_g\right\| \geq s\right) s^3 ds + \int_{s_d}^\infty \mathbb{P}\left(\left\|\widehat{\Sigma}_g - \Sigma_g\right\| \geq s\right) s^3 ds$$

For the first term, we may take the trivial upper bound of 1 to bound the integral as follows:

$$\int_0^{s_d} \mathbb{P}\left(\left\|\widehat{\Sigma}_g - \Sigma_g\right\| \geq s\right) s^3 ds \leq \int_0^{s_d} s^3 ds \leq \frac{1}{4}\max\{s_1^4, s_2^4\} = \frac{1}{4}\max\left\{\left(\frac{d}{C'n_g}\right)^2, \left(\frac{d}{C'n_g}\right)^4\right\}$$

For the second term, observe

$$\int_{s_d}^\infty \mathbb{P}\left(\left\|\widehat{\Sigma}_g - \Sigma_g\right\| \geq s\right) s^3 ds \leq \max\left\{\int_{s_1}^\infty \mathbb{P}\left(\left\|\widehat{\Sigma}_g - \Sigma_g\right\| \geq s\right) s^3 ds, \int_{s_2}^\infty \mathbb{P}\left(\left\|\widehat{\Sigma}_g - \Sigma_g\right\| \geq s\right) s^3 ds\right\}.$$

We then bound each integral:

$$\int_{s_1}^\infty \mathbb{P}\left(\left\|\widehat{\Sigma}_g - \Sigma_g\right\| \geq s\right) s^3 ds \leq \int_{s_1}^\infty s^3 \exp\left(\frac{-C'n_g s^2}{K_g^4 \|\Sigma_g\|^2}\right) ds = \frac{K_g^8(d+1)\exp(-d)\|\Sigma_g\|^4}{2C'^2 n_g^2}$$

$$\int_{s_2}^\infty \mathbb{P}\left(\left\|\widehat{\Sigma}_g - \Sigma_g\right\| \geq s\right) s^3 ds \leq \int_{s_2}^\infty s^3 \exp\left(\frac{-C'n_g s}{K_g^2 \|\Sigma_g\|}\right) ds = \frac{K_g^8\left(d^3 + 3d^2 + 6d + 6\right)\exp(-d)\|\Sigma_g\|^4}{C'^4 n_g^4}.$$

Therefore, in either case, we conclude that, for an absolute constant $C > 0$,

$$\mathbb{E}\left[\left\|\widehat{\Sigma}_g - \Sigma_g\right\|^4\right] \leq C K_g^8 \|\Sigma_g\|^4 \frac{d^2}{n_g^2}$$

as desired. ∎

**Lemma 5.** *Under Assumption 4, there exists a constant $C > 0$ such that*

$$\mathbb{E}\left[\operatorname{Tr}\left(\widehat{\Sigma}_g^2\right)\right] \leq \operatorname{Tr}\left(\Sigma_g^2\right) + \frac{CK_g^4}{n_g}\operatorname{Tr}\left(\Sigma_g\right)^2.$$

**Proof of Lemma 5**

For notational convenience, we write $X_i$ in place of $X_i^{(g)} \sim X|G = g$ throughout the proof. Applying linearity of the trace and expectation and expanding the empirical covariance matrix product, we have:

$$\mathbb{E}\left[\operatorname{Tr}\left(\widehat{\Sigma}_g^2\right)\right] = \frac{1}{n_g^2}\sum_{i=1}^{n_g}\operatorname{Tr}\left(\mathbb{E}[X_i X_i^\top X_i X_i^\top]\right) + \frac{1}{n_g^2}\sum_{i \neq j}\operatorname{Tr}\left(\mathbb{E}[X_i X_i^\top X_j X_j^\top]\right)$$

For the cross terms, by independence, for $i \neq j$,

$$\mathbb{E}\left[X_i X_i^\top X_j X_j^\top\right] = \mathbb{E}\left[X_i X_i^\top\right]\mathbb{E}\left[X_j X_j^\top\right] = \Sigma_g^2.$$

Thus,

$$\sum_{i \neq j}\operatorname{Tr}\left(\mathbb{E}[X_i X_i^\top X_j X_j^\top]\right) = n_g(n_g - 1)\operatorname{Tr}\left(\Sigma_g^2\right).$$

For the diagonal terms, interchanging the trace and expectation and applying invariance under cyclic permutations of the trace, we have

$$\operatorname{Tr}\left(\mathbb{E}\left[X_i X_i^\top X_i X_i^\top\right]\right) = \mathbb{E}\left[\|X_i\|_2^4\right].$$

By our subgaussianity assumption 4 and standard moment bounds for subgaussian vectors (see, e.g., [54]), we have, for some absolute constant $C > 0$:

$$\mathbb{E}\left[\|X_1\|^4\right] \leq C K_g^4 \operatorname{Tr}\left(\Sigma_g\right)^2.$$

Put the pieces together, we have

$$\mathbb{E}\left[\operatorname{Tr}\left(\widehat{\Sigma}_g^2\right)\right] \leq \operatorname{Tr}\left(\Sigma_g^2\right) + \frac{CK_g^4}{n_g}\operatorname{Tr}(\Sigma_g)^2$$

as desired. ∎

**Lemma 6** (Assouad's Lemma [59]). *Fix an integer $m \geq 1$ and define the $m$-dimensional hypercube $\Xi_m := \{-1, 1\}^m$. Let $\mathcal{P} = \{\mathbb{P}_\xi : \xi \in \Xi_m\}$ be a family of probability measures such that for each $\xi \in \Xi_m$ there is an associated parameter $\theta(\mathbb{P}_\xi)$ that is the target of interest. Also suppose that $|\mathcal{P}| = 2^m$. Take the loss function to be the squared $\ell_2$-distance. For $j \in [m]$, let $\xi^{(j)}$ denote the vector obtained by flipping the $j$-th coordinate of vector $\xi$, i.e., $\xi_{-j} = \xi^{(j)}_{-j}$, $\xi_j = -\xi^{(j)}_j$. Define the per-coordinate separation as*

$$\alpha := \inf_{j \in [m], \xi \in \Xi_m} \left\langle e_j, \theta(\mathbb{P}_\xi) - \theta\left(\mathbb{P}_{\xi^{(j)}}\right) \right\rangle^2.$$

*Then, for every estimator $\widehat{\theta}$, the minimax risk is lower bounded by*

$$\sup_{\mathbb{P}_\xi \in \mathcal{P}} \mathbb{E}_{\mathbb{P}_\xi} \left[ \left\| \widehat{\theta} - \theta(\mathbb{P}_\xi) \right\|_2^2 \right] \geq \frac{m}{2} \cdot \alpha \cdot \left( 1 - \sup_{\substack{\xi, \xi' \in \Xi_m \\ d_h(\xi, \xi') = 1}} \| \mathbb{P}_\xi, \mathbb{P}_{\xi'} \|_{TV} \right)$$

*where $d_h$ denotes the Hamming distance.*

### C.1 Proofs from Section 3

#### C.1.1 Proof of Proposition 1

Note that

$$\mathcal{R}_g(\beta) = \mathbb{E}_g \left[ (X^\top \beta - Y)^2 \right] = \mathbb{E}_g \left[ (X^\top(\beta - \beta_g) - \varepsilon_g)^2 \right] = \| \beta - \beta_g \|_{\Sigma_g}^2 + \sigma^2, \quad (23)$$

where the last equality follows from the fact that $\mathbb{E}[\varepsilon_g | X] = 0$. As a result, we have

$$\mathcal{R}_g(\beta) - \mathcal{R}_g(\beta_\lambda) = \| \beta - \beta_g \|_{\Sigma_g}^2 - \| \beta_\lambda - \beta_g \|_{\Sigma_g}^2 = \| \beta - \beta_\lambda \|_{\Sigma_g}^2 + 2(\beta - \beta_\lambda)^\top \Sigma_g (\beta_\lambda - \beta_g), \quad (24)$$

which completes the proof of (4b). To establish (4a), simply sum both sides of (4b) for the red and blue groups, weighted by $\lambda$ and $1 - \lambda$, respectively, and use the definition of $\beta_\lambda$.

### C.2 Proofs from Section 4

#### C.2.1 Proof of Theorem 1

**Upper bound**

We first establish that the error given in the statement of the theorem is indeed achieved by setting $\beta = \tilde{\beta}_\lambda$. Notice that, we have

$$\tilde{\beta}_\lambda = \beta_\lambda - \Sigma_\lambda^{-1} \left( \lambda \Sigma_r (\beta_r - \widehat{\beta}_r) + (1 - \lambda) \Sigma_b (\beta_b - \widehat{\beta}_b) \right).$$

Substituting $\widehat{\beta}_g = \widehat{\Sigma}_g^{-1} \widehat{\nu}_b$ and using the decomposition

$$\widehat{\nu}_g = \widehat{\Sigma}_g \beta_g + \frac{1}{n_g} \sum_{i=1}^{n_g} X_i^{(g)} \varepsilon_i^{(g)},$$

we obtain

$$\tilde{\beta}_\lambda - \beta_\lambda = \Sigma_\lambda^{-1} \left( \lambda \Sigma_r \widehat{\Sigma}_r^{-1} \cdot \frac{1}{n_r} \sum_{i=1}^{n_r} X_i^{(r)} \varepsilon_i^{(r)} + (1 - \lambda) \Sigma_b \widehat{\Sigma}_b \cdot \frac{1}{n_b} \sum_{i=1}^{n_b} X_i^{(b)} \varepsilon_i^{(b)} \right).$$

Let $A := \Sigma_\lambda^{-1} \Sigma_g \Sigma_\lambda^{-1}$. By the mean-zero and independence properties of the noise, the cross terms cancel in the squared norm, which yields

$$\mathbb{E} \left[ \left\| \tilde{\beta}_\lambda - \beta_\lambda \right\|_{\Sigma_g}^2 \right] = \lambda^2 \cdot \frac{1}{n_r^2} \mathbb{E} \left[ \left\| \Sigma_r \widehat{\Sigma}_r^{-1} \sum_{i=1}^{n_r} X_i^{(r)} \varepsilon_i^{(r)} \right\|_A^2 \right] + (1 - \lambda)^2 \cdot \frac{1}{n_b^2} \mathbb{E} \left[ \left\| \Sigma_b \widehat{\Sigma}_b^{-1} \sum_{i=1}^{n_b} X_i^{(b)} \varepsilon_i^{(b)} \right\|_A^2 \right]$$

Using the independence and mean-zero property of $\varepsilon_i^{(g)}$ and applying the trace identity for the expectation of quadratic forms, we obtain

$$\mathbb{E}\left[\left\|\Sigma_g \widehat{\Sigma}_r^{-1} \sum_{i=1}^{n_r} X_i^{(r)} \varepsilon_i^{(r)}\right\|_A^2\right] = \sigma^2 n_r \cdot \mathbb{E}\left[\text{Tr}\left(\widehat{\Sigma}_r^{-1} \Sigma_r A \Sigma_r\right)\right].$$

Applying the same procedure for the equivalent term for group $b$ and substituting completes the proof.

**Lower bound**

Next, we establish the minimax optimality. In particular, we aim to show

$$\sup_{P \in \mathcal{P}_{\text{linear}}(P_x^r, P_X^b, \sigma^2)} \mathbb{E}\left[\|\beta - \beta_\lambda\|_{\Sigma_g}^2\right]$$

$$\geq \lambda^2 \frac{\sigma^2}{n_r} \mathbb{E}\left[\text{Tr}\left(\Sigma_g \Sigma_\lambda^{-1} \Sigma_r \widehat{\Sigma}_r^{-1} \Sigma_r \Sigma_\lambda^{-1}\right)\right] + (1-\lambda)^2 \frac{\sigma^2}{n_b} \mathbb{E}\left[\text{Tr}\left(\Sigma_g \Sigma_\lambda^{-1} \Sigma_b \widehat{\Sigma}_b^{-1} \Sigma_b \Sigma_\lambda^{-1}\right)\right],$$

for any $\beta$. We derive this lower bound by the Bayes risk under an appropriate prior. More specifically, suppose we choose a prior over groups' regressor $\beta_r$ and $\beta_b$. Then, the Bayes risk provides a valid lower bound:

$$\inf_\beta \sup_{\beta_\lambda} \mathbb{E}\left[\|\beta - \beta_\lambda\|_{\Sigma_g}^2\right] \geq \mathbb{E}\left[\left\|\widehat{\beta}_\lambda^* - \beta_\lambda\right\|_{\Sigma_g}\right], \tag{25}$$

where $\widehat{\beta}_\lambda^*$ denotes the Bayes estimator under a prior specified below.

Let $\tau > 0$ and suppose that the group-specific parameters follow independent Gaussian priors:

$$\beta_r \sim \mathcal{N}(0, \tau^2 I_d), \quad \beta_b \sim \mathcal{N}(0, \tau^2 I_d).$$

Then, the induced prior on $\beta_\lambda$ is also Gaussian:

$$\beta_\lambda \sim \mathcal{N}\left(0, \Sigma_\lambda^{-1}\left(\lambda^2 \tau^2 \Sigma_r^2 + (1-\lambda)^2 \tau^2 \Sigma_b^2\right)\Sigma_\lambda^{-1}\right).$$

Under the linear model $Y = X^\top \beta_g + \varepsilon_g$ with i.i.d noise $\varepsilon_g \sim \mathcal{N}(0, \sigma^2)$, the posterior distribution for each parameter $\beta_g | \left\{X_i^{(g)}, Y_i^{(g)}\right\}$ is Gaussian:

$$\beta_g | \left\{X_i^{(g)}, Y_i^{(g)}\right\} \sim \mathcal{N}\left(\mu_g(\tau), V_g(\tau)\right)$$

with posterior mean and variance:

$$M_g(\tau) := \widehat{\Sigma}_g + \frac{\sigma^2}{n_g \tau^2} I_d, \quad \mu_g(\tau) := M_g(\tau)^{-1} \widehat{\nu}_g, \quad V_g(\tau) := \frac{\sigma^2}{n_g} M_g(\tau)^{-1}.$$

Since $\beta_\lambda$ is a deterministic linear combination of $\beta_r$ and $\beta_b$, its posterior is also Gaussian:

$$\beta_\lambda \mid \mathcal{S}_n \sim \mathcal{N}\left(\mu_\lambda(\tau), V_\lambda(\tau)\right)$$

where:

$$\mu_\lambda(\tau) := \Sigma_\lambda^{-1}\left(\lambda \Sigma_r \mu_r(\tau) + (1-\lambda)\Sigma_b \mu_b(\tau)\right), \quad V_\lambda := \Sigma_\lambda^{-1}\left(\lambda^2 \Sigma_r V_r(\tau)\Sigma_r + (1-\lambda)^2 \Sigma_b V_b(\tau)\Sigma_b\right)\Sigma_\lambda^{-1}.$$

Since the Bayes estimator under squared loss is the posterior mean, the Bayes risk in the $\Sigma_g$-norm is:

$$\mathbb{E}\left[\left\|\widehat{\beta}_\lambda^* - \beta_\lambda\right\|_{\Sigma_g}\right] = \mathbb{E}\left[\text{Tr}\left(\Sigma_g V_\lambda(\tau)\right)\right]$$

To obtain a lower bound, we consider the limit $\tau \to \infty$, which corresponds to an uninformative prior. In this limit:

$$\mu_g(\tau) \to \widehat{\Sigma}^{-1} \widehat{\nu}_g, \quad V_g(\tau) \to \frac{\sigma^2}{n_g} \widehat{\Sigma}_g^{-1}.$$

Thus, $V_\lambda(\tau)$ increases in the Loewner order as $\tau \to \infty$, and by monotone convergence,

$$\lim_{\tau \to \infty} \mathbb{E}\left[\text{Tr}\left(\Sigma_g V_\lambda(\tau)\right)\right] = \lambda^2 \frac{\sigma^2}{n_r} \mathbb{E}\,\text{Tr}\left(\Sigma_g \Sigma_\lambda^{-1} \Sigma_r \widehat{\Sigma}_r^{-1} \Sigma_r \Sigma_\lambda^{-1}\right) + (1-\lambda)^2 \frac{\sigma^2}{n_b} \mathbb{E}\,\text{Tr}\left(\Sigma_g \Sigma_\lambda^{-1} \Sigma_b \widehat{\Sigma}_b^{-1} \Sigma_b \Sigma_\lambda^{-1}\right).$$

The desired minimax lower bound follows from substitution into the inequality (25).

### C.2.2 Proof of Corollary 2

Under the group-wise spherical covariance assumption, we have

$$\Sigma_\lambda^{-1} = \left(\lambda\rho_r^2 + (1-\lambda)\rho_b^2\right)^{-1} I_d.$$

Substituting and applying linearity of the trace yields

$$\mathbb{E}\left[\left\|\tilde{\beta}_\lambda - \beta_\lambda\right\|_{\Sigma_g}^2\right] \le \lambda^2 \frac{\sigma^2}{n_r} \frac{\rho_g^2\rho_r^4}{(\lambda\rho_r^2 + (1-\lambda)\rho_b^2)^2} \mathbb{E}\left[\mathrm{Tr}\left(\widehat{\Sigma}_r^{-1}\right)\right] + (1-\lambda)^2 \frac{\sigma^2}{n_b} \frac{\rho_g^2\rho_b^4}{(\lambda\rho_r^2 + (1-\lambda)\rho_b^2)^2} \mathbb{E}\left[\mathrm{Tr}\left(\widehat{\Sigma}_b^{-1}\right)\right].$$

$$(26)$$

Let $\tilde{X}_i^{(g)}$ denote the whitened covariate vectors, i.e., $\tilde{X}_i^{(g)} = \rho_g^{1/2} X_i^{(g)}$ such that $\mathbb{E}\left[\tilde{X}_i^{(g)}\tilde{X}_i^{(g)\top}\right] = I_d.$

Also observe

$$\mathrm{Tr}\left(\tilde{\Sigma}_g^{-1}\right) \le d \cdot \lambda_{\min}\left(\tilde{\Sigma}_g\right)^{-1}.$$

By Corollary 4 of [43],

$$\mathbb{E}\left[\lambda_{\min}\left(\tilde{\Sigma}_g\right)^{-1}\right] \le 2C_g'.$$

Then, noting $\mathrm{Tr}\left(\tilde{\Sigma}_g\right) = \rho_g^2 \mathrm{Tr}\left(\widehat{\Sigma}_g\right)$, we reach the claim by substituting the resulting bound

$$\mathbb{E}\left[\mathrm{Tr}\left(\widehat{\Sigma}_g^{-1}\right)\right] \le 2C_g'\rho_g^{-2}d \tag{27}$$

into (26).

To show the second half of the claim, consider subgaussian $X \in \mathbb{R}^d$ satisfying 4 with parameter $K_g$ and with covariance $\rho_g^2 I_d$. Each coordinate satisfies $\|X_i\|_{\psi_2} \le K_g\rho_g$ and it follows that there exists an absolute constant $\zeta_0$ such that $\mathbb{E}\left[X_i^4\right] \le \zeta_0 K_g^4\rho_g^4$. By Cauchy-Schwarz, for $i \ne j$, $\mathbb{E}\left[X_i^2 X_j\right] \le \zeta_0 K_g^4\rho_g^4$. Hence, there exists a constant $\zeta > 0$ such that

$$\mathbb{E}\left[\|X\|^4\right] = d \cdot \mathbb{E}\left[X_1^4\right] + 2\binom{d}{2} \cdot \mathbb{E}\left[X_1^2 X_2^2\right] \le \zeta K_g^4\rho_g^4.$$

Then, by whitening and applying Mourtada [43, Theorem 3], we replace (27) with the following bound:

$$\mathbb{E}\left[\mathrm{Tr}\left(\widehat{\Sigma}_g^{-1}\right)\right] \le \rho_g^{-2}\left(d + \frac{8C_g'\zeta\rho_g^8 d^2}{n_g}\right)$$

and the claim follows. ∎

### C.3 Proofs from Section 5

#### C.3.1 Proof of Proposition 2

From Lemma 2, the excess risk decomposes as:

$$\mathbb{E}\left[\left\|\widehat{\beta}_\lambda - \beta_\lambda\right\|_{\Sigma_g}^2\right] \le \lambda^2 \frac{\sigma^2}{n_r^2}\mathbb{E}\left[\sum_{i=1}^{n_r}\left\|\tilde{X}_i\right\|_{\tilde{\Sigma}_g^{-2}}^2\right] + (1-\lambda)^2 \frac{\sigma^2}{n_b^2}\mathbb{E}\left[\sum_{i=1}^{n_b}\left\|\tilde{X}_i\right\|_{\tilde{\Sigma}_g^{-2}}^2\right]$$

$$+ \mathbb{E}\left[\left\|\lambda\Sigma_g^{-1/2}\widehat{\Sigma}_r(\beta_r - \beta_\lambda) + (1-\lambda)\Sigma_g^{-1/2}\widehat{\Sigma}_b(\beta_b - \beta_\lambda)\right\|_{\tilde{\Sigma}_g^{-2}}^2\right]$$

where we define:

$$\tilde{X}_i = \Sigma_g^{-1/2}X_i \quad \text{and} \quad \tilde{\Sigma}_g = \Sigma_g^{-1/2}\widehat{\Sigma}_\lambda \Sigma_g^{-1/2}.$$

We apply the identity:

$$\sum_{i=1}^n \|Ax_i\|^2 = \mathrm{Tr}\left(A^\top A \sum_{i=1}^n x_i x_i^\top\right)$$

with $A = \tilde{\Sigma}_g^{-2}$ and $x_i = \tilde{X}_i$. This gives:

$$\frac{\sigma^2}{n_r^2}\mathbb{E}\left[\sum_{i=1}^{n_r}\left\|\tilde{X}_i\right\|^2_{\tilde{\Sigma}_g^{-2}}\right] = \frac{\sigma^2}{n_r}\mathbb{E}\,\mathrm{Tr}\left(\widehat{\Sigma}_\lambda^{-1}\Sigma_g\widehat{\Sigma}_\lambda^{-1}\widehat{\Sigma}_r\right), \quad \frac{\sigma^2}{n_b^2}\mathbb{E}\left[\sum_{i=1}^{n_b}\left\|\tilde{X}_i\right\|^2_{\tilde{\Sigma}_g^{-2}}\right] = \frac{\sigma^2}{n_b}\mathbb{E}\,\mathrm{Tr}\left(\widehat{\Sigma}_\lambda^{-1}\Sigma_g\widehat{\Sigma}_\lambda^{-1}\widehat{\Sigma}_b\right)$$

For the final bias term, observe that:

$$\left\|\lambda\Sigma_g^{-1/2}\widehat{\Sigma}_r(\beta_r - \beta_\lambda) + (1-\lambda)\Sigma_g^{-1/2}\widehat{\Sigma}_b(\beta_b - \beta_\lambda)\right\|^2_{\tilde{\Sigma}_g^{-2}} = \left\|\Sigma_g^{1/2}\widehat{\Sigma}_\lambda^{-1}\left(\lambda\widehat{\Sigma}_r(\beta_r - \beta_\lambda) + (1-\lambda)\widehat{\Sigma}_b(\beta_b - \beta_\lambda)\right)\right\|^2,$$

which gives the first characterization of the bias term. To see the other representation, note that

$$\lambda\Sigma_r(\beta_r - \beta_\lambda) + (1-\lambda)\Sigma_b(\beta_b - \beta_\lambda) = 0.$$

Therefore, we have

$$\lambda\widehat{\Sigma}_r(\beta_r - \beta_\lambda) + (1-\lambda)\widehat{\Sigma}_b(\beta_b - \beta_\lambda) = \lambda\left(\widehat{\Sigma}_r - \Sigma_r\right)(\beta_r - \beta_\lambda) + (1-\lambda)\left(\widehat{\Sigma}_b - \Sigma_b\right)(\beta_b - \beta_\lambda).$$

Then, we can rearrange to see that

$$\widehat{\Sigma}_\lambda^{-1}\left(\lambda\left(\widehat{\Sigma}_r - \Sigma_r\right)(\beta_r - \beta_\lambda) + (1-\lambda)\left(\widehat{\Sigma}_b - \Sigma_b\right)(\beta_b - \beta_\lambda)\right) \tag{28}$$

$$= \lambda\widehat{\Sigma}_\lambda^{-1}\widehat{\Sigma}_r\beta_r + (1-\lambda)\widehat{\Sigma}_\lambda^{-1}\widehat{\Sigma}_b\beta_b - \beta_\lambda$$

$$= \lambda\left(\widehat{\Sigma}_\lambda^{-1}\widehat{\Sigma}_r - \Sigma_\lambda^{-1}\Sigma_r\right)\beta_r + (1-\lambda)\left(\widehat{\Sigma}_\lambda^{-1}\widehat{\Sigma}_b - \Sigma_\lambda^{-1}\Sigma_b\right)\beta_b. \tag{29}$$

Next, note that, we have

$$\lambda\left(\widehat{\Sigma}_\lambda^{-1}\widehat{\Sigma}_r - \Sigma_\lambda^{-1}\Sigma_r\right) = \left(I_d + \frac{1-\lambda}{\lambda}\widehat{\Sigma}_r^{-1}\widehat{\Sigma}_b\right)^{-1} - \left(I_d + \frac{1-\lambda}{\lambda}\Sigma_r^{-1}\Sigma_b\right)^{-1}. \tag{30}$$

Similarly, we can write

$$(1-\lambda)\left(\widehat{\Sigma}_\lambda^{-1}\widehat{\Sigma}_b - \Sigma_\lambda^{-1}\Sigma_b\right) = \left(I_d + \frac{\lambda}{1-\lambda}\widehat{\Sigma}_b^{-1}\widehat{\Sigma}_r\right)^{-1} - \left(I_d + \frac{\lambda}{1-\lambda}\Sigma_b^{-1}\Sigma_r\right)^{-1}. \tag{31}$$

Next, using the Woodbury matrix identity, we have $(I_d + X)^{-1} = I_d - (I_d + X^{-1})^{-1}$ for any invertible $d \times d$ matrix $X$. Therefore, we can recast (31) as

$$(1-\lambda)\left(\widehat{\Sigma}_\lambda^{-1}\widehat{\Sigma}_b - \Sigma_\lambda^{-1}\Sigma_b\right) = -\left(I_d + \frac{1-\lambda}{\lambda}\widehat{\Sigma}_r^{-1}\widehat{\Sigma}_b\right)^{-1} + \left(I_d + \frac{1-\lambda}{\lambda}\Sigma_r^{-1}\Sigma_b\right)^{-1}. \tag{32}$$

Plugging (30) and (32) into (29) completes the proof. ∎

### C.3.2   Proof of Theorem 2

**Upper bound**

We show a slightly more general result for the upper bound. In fact, we relax the (14) to

$$\rho_g^2 I_d \preceq \Sigma_g \preceq \mathrm{P}_g^2 I_d, \tag{33}$$

and also drop the assumption $n_g \geq K_g^4 d$, and show the upper bound

$$\mathcal{V}_g(\lambda) \lesssim \frac{\mathrm{P}_g^2\sigma^2}{(\lambda\rho_r^2/C_r' + (1-\lambda)\rho_b^2/C_b')^2}\left[\lambda^2\mathrm{P}_r^2\left(\frac{d}{n_r} + \frac{K_r^2 d^{3/2}}{n_r^{3/2}}\right) + (1-\lambda)^2\mathrm{P}_b^2\left(\frac{d}{n_b} + \frac{K_b^2 d^{3/2}}{n_b^{3/2}}\right)\right]. \tag{34}$$

To do so, first apply Proposition 2 under $\Sigma_g \preceq \mathrm{P}_g^2 I_d$ to obtain the following bound:

$$\mathcal{V}_g(\lambda) \leq \mathrm{P}_g^2\left(\lambda^2\frac{\sigma^2}{n_r}\mathbb{E}\left[\mathrm{Tr}\left(\widehat{\Sigma}_\lambda^{-2}\widehat{\Sigma}_r\right)\right] + (1-\lambda)^2\frac{\sigma^2}{n_b}\mathbb{E}\left[\mathrm{Tr}\left(\widehat{\Sigma}_\lambda^{-2}\widehat{\Sigma}_b\right)\right]\right)$$

Next, by Cauchy-Schwarz on the trace inner product and again on the expectation:

$$\mathbb{E}\left[\mathrm{Tr}\left(\widehat{\Sigma}_\lambda^{-2}\widehat{\Sigma}_g\right)\right] \leq \mathbb{E}\left[\sqrt{\mathrm{Tr}\left(\widehat{\Sigma}_\lambda^{-4}\right)}\sqrt{\mathrm{Tr}\left(\widehat{\Sigma}_g^2\right)}\right] \leq \sqrt{\mathbb{E}\left[\mathrm{Tr}\left(\widehat{\Sigma}_\lambda^{-4}\right)\right]\mathbb{E}\left[\mathrm{Tr}\left(\widehat{\Sigma}_g^2\right)\right]}.$$

Under Assumption 4 that $X|G = g$ is $K_g$-subgaussian, by Lemma 5,

$$\mathbb{E}\left[\text{Tr}\left(\widehat{\Sigma}_g^2\right)\right] \leq \text{Tr}\left(\Sigma_g^2\right) + \frac{CK_g^4}{n_g}\text{Tr}\left(\Sigma_g\right)^2 \lesssim \text{P}_g^4\left(d + \frac{K_g^4 d^2}{n_g}\right) \tag{35}$$

where the second inequality uses that $\Sigma_g \preceq \text{P}_g^2 I_d$. Next, by Weyl's inequality,

$$\lambda_{\min}\left(\widehat{\Sigma}_\lambda\right) \geq \lambda \cdot \lambda_{\min}\left(\widehat{\Sigma}_r\right) + (1 - \lambda) \cdot \lambda_{\min}\left(\widehat{\Sigma}_g\right). \tag{36}$$

Therefore, we have

$$\mathbb{E}\left[\lambda_{\min}\left(\widehat{\Sigma}_\lambda\right)^{-4}\right] \leq \min\left\{\lambda^{-4}\mathbb{E}\left[\lambda_{\min}\left(\widehat{\Sigma}_r\right)^{-4}\right], (1 - \lambda)^{-4}\mathbb{E}\left[\lambda_{\min}\left(\widehat{\Sigma}_b\right)^{-4}\right]\right\} \tag{37}$$

Now, by Lemma 3, we have

$$\sqrt{\mathbb{E}\left[\text{Tr}\left(\widehat{\Sigma}_\lambda^{-4}\right)\right]} \leq \sqrt{2}\sqrt{d}\min\left\{\lambda^{-2}C_r'^2\rho_r^{-4}, (1 - \lambda)^{-2}C_b'^2\rho_b^{-4}\right\} \tag{38}$$

Combining the bounds (38) and (35) and using $\min\{1/a, 1/b\} \leq 2/(a + b)$, we reach the bound:

$$\mathbb{E}\left[\text{Tr}\left(\widehat{\Sigma}_\lambda^{-2}\widehat{\Sigma}_g\right)\right] \lesssim \text{P}_g^2 \frac{1}{\frac{\lambda^2}{C_r'^2}\rho_r^4 + \frac{(1-\lambda)^2}{C_b'^2}\rho_b^4}\left(d + \frac{K_g^2 d^{3/2}}{\sqrt{n_g}}\right) \tag{39}$$

$$\lesssim \text{P}_g^2 \frac{1}{(\lambda\rho_r^2/C_r' + (1-\lambda)\rho_b^2/C_b')^2}\left(d + \frac{K_g^2 d^{3/2}}{\sqrt{n_g}}\right) \tag{40}$$

As a result, the variance terms satisfy the following bound:

$$\text{P}_g^2\left(\lambda^2\frac{\sigma^2}{n_r}\mathbb{E}\left[\text{Tr}\left(\widehat{\Sigma}_\lambda^{-2}\widehat{\Sigma}_r\right)\right] + (1-\lambda)^2\frac{\sigma^2}{n_b}\mathbb{E}\left[\text{Tr}\left(\widehat{\Sigma}_\lambda^{-2}\widehat{\Sigma}_b\right)\right]\right)$$

$$\lesssim \frac{\text{P}_g^2\sigma^2}{(\lambda\rho_r^2/C_r' + (1-\lambda)\rho_b^2/C_b')^2}\left[\lambda^2\text{P}_r^2\left(\frac{d}{n_r} + \frac{K_r^2 d^{3/2}}{n_r^{3/2}}\right) + (1-\lambda)^2\text{P}_b^2\left(\frac{d}{n_b} + \frac{K_b^2 d^{3/2}}{n_b^{3/2}}\right)\right].$$

Under the assumption $n_g \geq K_g^4 d$, the term $K_g^2 d^{3/2}/n_g^{3/2}$ is bounded by $d/n_g$ which completes the proof of the upper bound.

**Lower bound**

We use the Assouad's Lemma (Lemma 6). Here are the steps:

**Perturbed parameter structure:** For $g \in \{r, b\}$, set

$$h_g^2 = \frac{\sigma^2}{4n_g\rho_g^2}, \quad \beta_g^{(\xi)} := h_g\xi_g$$

where $\xi_g \in \Xi^d$. The condition on $n_g$ ensures $\|\beta_g^{(\xi)}\| \leq B$.

Consider the $2d$-dimensional hybercube $\Xi := \Xi^d \times \Xi^d$. For $\xi := (\xi_r, \xi_b) \in \Xi$, define by $\mathbb{P}^{(\xi)}$ the joint law of the data with mean $\beta^{(\xi)} := (\beta_r^{(\xi_r)}, \beta_b^{(\xi_b)})$.

**KL divergence bound:** The following result provides an upper bound on the KL divergence between the distributions under parameters $\xi, \xi' \in \Xi$ differing in only a single coordinate.

**Claim 1.** *Let $\xi$ and $\xi'$ differ only in coordinate $i \in [d]$ (i.e., a coordinate corresponding to group $r$) where $\xi_i = -\xi_i'$. Then*

$$D_{KL}(\mathbb{P}^{(\xi)} \parallel \mathbb{P}^{(\xi')}) \leq \frac{2h_r^2\rho_r^2}{\sigma^2}$$

*Proof of Claim 1.* Take $\xi$ and $\xi'$ to be neighbors that differ only in the $i$-th coordinate of $\xi_r$, i.e., $\xi_{r,i} = 1, \xi_{r,i} = -1$ and $\xi_{r,-i} = \xi'_{r,-i}$. Recalling well-known results of the KL divergence of multivariate Gaussians (see, e.g., [61]), conditional on observing a single observation $X_i$, the KL divergence between $\mathbb{P}_{\xi_r} = \mathcal{N}(h_r X_i, \sigma^2), \mathbb{P}_{\xi'_r} = \mathcal{N}(-h_r X_i, \sigma^2)$ is given by

$$D_{\mathrm{KL}}(\mathbb{P}_{\xi_r} \parallel \mathbb{P}_{\xi'_r} \mid X_i) = \frac{2h_r^2 X_i^2}{\sigma^2}.$$

Then, for $X_i \sim \mathcal{N}(0, \rho_r^2)$ with $\mathbb{E}[X_i^2] = \rho_r^2$, we observe that

$$D_{\mathrm{KL}}(\mathbb{P}_{\xi_r} \parallel \mathbb{P}_{\xi'_r}) = \frac{2h_r^2 \rho_r^2}{\sigma^2}.$$

$\square$

By Claim 1, over $n_r$ samples,

$$D_{\mathrm{KL}}(\mathbb{P}^{(\xi) \otimes n_r} \parallel \mathbb{P}^{(\xi') \otimes n_r}) \leq \frac{2h_r^2 \rho_r^2 n_r}{\sigma^2} = \frac{1}{2}.$$

By Pinsker's inequality,

$$\|\mathbb{P}_\xi, \mathbb{P}_{\xi'}\|_{\mathrm{TV}} \leq \frac{1}{2}.$$

**Parameter separation:** Let $\xi$ and $\xi'$ differ only in the same coordinate $i \in [d]$ as in Claim 1 where $\xi_i = -\xi'_i$. Then, we can express

$$\langle e_i, \beta_\lambda^{(\xi)} - \beta_\lambda^{(\xi')} \rangle = \frac{\lambda \rho_r^2 (h_r - (-h_r))}{\rho_\lambda^2} = \frac{2\lambda \rho_r^2 h_r}{\rho_\lambda^2}, \tag{41}$$

with

$$\rho_\lambda^2 = \lambda \rho_r^2 + (1 - \lambda)\rho_b^2. \tag{42}$$

After an analogous computation for a single-coordinate group $b$ perturbation,

$$\langle e_i, \beta_\lambda^{(\xi)} - \beta_\lambda^{(\xi')} \rangle = \frac{2(1 - \lambda)\rho_b^2 h_b}{\rho_\lambda^2}. \tag{43}$$

Substitute the definitions of $h_r$ and $h_b$ into (41) and (43), respectively, and define

$$\alpha_r^2 := \frac{\sigma^2 \lambda^2 \rho_r^2}{n_r \rho_\lambda^4}, \quad \alpha_b^2 := \frac{\sigma^2 (1 - \lambda)^2 \rho_b^2}{n_b \rho_\lambda^4}.$$

**Assouad's Lemma:** Over a $2d$-dimensional signed hypercube, an application of Assouad's lemma (as stated in 6) and the fact $\max\{a, b\} \geq \frac{1}{2}(a + b)$ yields the bound

$$\inf_\beta \sup_{\xi \in \Xi} \mathbb{E}_\xi \left[ \left\| \beta_\lambda - \beta_\lambda^{(\xi)} \right\|_2^2 \right] \geq \frac{1}{16} \frac{\sigma^2 d}{\rho_\lambda^4} \left( \frac{\lambda^2 \rho_r^2}{n_r} + \frac{(1 - \lambda)^2 \rho_b^2}{n_b} \right).$$

and the final bound follows by observing

$$\|\beta_\lambda - \beta_\lambda\|_{\Sigma_g}^2 \geq \rho_g^2 \|\beta_\lambda - \beta_\lambda\|_2^2. \quad \blacksquare$$

### C.3.3 Proof of Theorem 3

**Upper Bound**

We again show a slightly more general result for the upper bound. In fact, we relax the (14) to

$$\rho_g^2 I_d \preceq \Sigma_g \preceq \mathrm{P}_g^2 I_d, \tag{44}$$

and show the upper bound

$$\mathcal{B}_g(\lambda) \lesssim \frac{\lambda^2 (1 - \lambda)^2 \mathrm{P}_g^2 \mathrm{P}_r^4 \mathrm{P}_b^4}{(\lambda \rho_r^2 / C'_r + (1 - \lambda)\rho_b^2 / C'_b)^2 \rho_\lambda^4} \cdot d \cdot \left( \frac{K_r^4}{n_r} + \frac{K_b^4}{n_b} \right) \|\beta_r - \beta_b\|^2, \tag{45}$$

with
$$\rho_\lambda^2 = \lambda\rho_r^2 + (1-\lambda)\rho_b^2. \tag{46}$$

To do so, first apply Proposition 2 under $\Sigma_g \preceq \mathrm{P}_g^2 I_d$ to obtain the following bound:

$$\mathcal{B}_g(\lambda) \le \mathrm{P}_g^2 \mathbb{E}\left[\left\|\widehat{\Sigma}_\lambda^{-1}\left(\lambda\widehat{\Sigma}_r(\beta_r - \beta_\lambda) + (1-\lambda)\widehat{\Sigma}_b(\beta_b - \beta_\lambda)\right)\right\|^2\right]. \tag{47}$$

Next, observe that, be definition of $\beta_\lambda$,

$$\lambda\Sigma_r(\beta_r - \beta_\lambda) + (1-\lambda)\Sigma_b(\beta_b - \beta_\lambda) = 0.$$

Subtracting this identity from its empirical counterpart therefore yields,

$$\tilde{A} := \lambda\widehat{\Sigma}_r(\beta_r - \beta_\lambda) + (1-\lambda)\widehat{\Sigma}_b(\beta_b - \beta_\lambda) = \lambda(\widehat{\Sigma}_r - \Sigma_r)(\beta_r - \beta_\lambda) + (1-\lambda)(\widehat{\Sigma}_b - \Sigma_b)(\beta_b - \beta_\lambda). \tag{48}$$

By triangle inequality,

$$\left\|\tilde{A}\right\| \le \lambda\left\|\widehat{\Sigma}_r - \Sigma_r\right\|\|\beta_r - \beta_\lambda\| + (1-\lambda)\left\|\widehat{\Sigma}_b - \Sigma_b\right\|\|\beta_b - \beta_\lambda\|. \tag{49}$$

Next, notice that we can express

$$\beta_r - \beta_\lambda = (1-\lambda)\,\Sigma_\lambda^{-1}\Sigma_b(\beta_r - \beta_b) \tag{50a}$$
$$\beta_b - \beta_\lambda = \lambda\,\Sigma_\lambda^{-1}\Sigma_r(\beta_b - \beta_r). \tag{50b}$$

Substituting into (50a) and (50b) into (49) and applying the assumptions $\|\Sigma_g\| \le \mathrm{P}_g^2$ and $\lambda_{\min}(\Sigma_\lambda) \ge \rho_\lambda^2$, we reach

$$\left\|\tilde{A}\right\| \le \frac{\lambda(1-\lambda)}{\rho_\lambda^2}\left(\mathrm{P}_b^2\left\|\widehat{\Sigma}_r - \Sigma_r\right\| + \mathrm{P}_r^2\left\|\widehat{\Sigma}_b - \Sigma_b\right\|\right)\|\beta_r - \beta_b\|. \tag{51}$$

Raising (51) to the fourth power and using the inequality $(a+b)^4 \le 8(a^4+b^4)$,

$$\left\|\tilde{A}\right\|^4 \le \frac{8\lambda^4(1-\lambda)^4}{\rho_\lambda^8}\left(\mathrm{P}_b^8\left\|\widehat{\Sigma}_r - \Sigma_r\right\|^4 + \mathrm{P}_r^8\left\|\widehat{\Sigma}_b - \Sigma_b\right\|^4\right)\|\beta_r - \beta_b\|^4.$$

Taking an expectation and square root successively then using that $\sqrt{a+b} \le \sqrt{a} + \sqrt{b}$,

$$\sqrt{\mathbb{E}\left[\left\|\tilde{A}\right\|^4\right]} \le \frac{2\sqrt{2}\lambda^2(1-\lambda)^2}{\rho_\lambda^4}\left(\mathrm{P}_b^4\mathbb{E}\left[\left\|\widehat{\Sigma}_r - \Sigma_r\right\|^4\right]^{1/2} + \mathrm{P}_r^4\mathbb{E}\left[\left\|\widehat{\Sigma}_b - \Sigma_b\right\|^4\right]^{1/2}\right)\|\beta_r - \beta_b\|^2.$$

Applying the subgaussian covariance estimation bound Lemma 4 to each term $\mathbb{E}\left[\left\|\widehat{\Sigma}_g - \Sigma_g\right\|^4\right]$, we obtain

$$\sqrt{\mathbb{E}\left[\left\|\tilde{A}\right\|^4\right]} \le \frac{2\sqrt{2}C\lambda^2(1-\lambda)^2}{\rho_\lambda^4}\mathrm{P}_r^4\mathrm{P}_b^4 d\left(\frac{K_r^4}{n_r} + \frac{K_b^4}{n_b}\right)\|\beta_r - \beta_b\|^2. \tag{52}$$

Next, recall the property

$$\left\|\widehat{\Sigma}_\lambda^{-1}\tilde{A}\right\|^2 \le \lambda_{\min}\left(\widehat{\Sigma}_\lambda\right)^{-2}\left\|\tilde{A}\right\|^2. \tag{53}$$

By Cauchy-Schwarz and (53),

$$\mathbb{E}\left[\left\|\widehat{\Sigma}_\lambda^{-1}\tilde{A}\right\|^2\right] \le \sqrt{\mathbb{E}\left[\lambda_{\min}\left(\widehat{\Sigma}_\lambda\right)^{-4}\right]}\sqrt{\mathbb{E}\left\|\tilde{A}\right\|^4}.$$

By Weyl's inequality, convexity, and Lemma 3 as in the preceding analysis of the variance term, we have

$$\sqrt{\mathbb{E}\left[\lambda_{\min}\left(\widehat{\Sigma}_\lambda\right)^{-4}\right]} \lesssim \frac{1}{(\lambda\rho_r^2/C_r' + (1-\lambda)\rho_b^2/C_b')^2}. \tag{54}$$

Combining the bounds (52) and (54), we conclude the proof of the upper bound on the bias term.

**Lower bound**

Fix $\rho_r^2, \rho_b^2 > 0$ and $\lambda \in (0,1)$. For each coordinate $i \in [d]$, set

$$v := \frac{\beta_r - \beta_b}{\|\beta_r - \beta_b\|_2}, \tag{55a}$$

$$u_i = \begin{cases} \frac{e_i + v}{\|e_i + v\|_2}, & \text{if } e_i^\top v \geq 0, \\ \frac{e_i - v}{\|e_i - v\|_2}, & \text{if } e_i^\top v \leq 0. \end{cases} \tag{55b}$$

**Perturbed covariance structure for group $r$:** For a Rademacher vector $\xi \in \{-1, 1\}^d$, set

$$\Sigma_r^{(\xi)} := \rho_r^2 I_d + h_r \sum_{i=1}^d \xi_i u_i u_i^\top, \quad \Sigma_b := \rho_b^2 I_d$$

where the group $r$ perturbation level $h_r$ is given by

$$h_r = \frac{2\rho_r^2}{5\sqrt{n_r}} \leq \frac{\rho_r^2}{10d}, \tag{56}$$

where the inequality follows from $n_r \geq 16d^2$. Hence, for every $\xi$,

$$0.9 \cdot \rho_r^2 I_d \preceq \Sigma_r^{(\xi)} \preceq 1.1\rho_r^2 I_d \tag{57}$$

for $d \geq 1$.

**KL diverence bound:** Define $\mathbb{P}^{(\xi)} := \mathcal{N}(0, \Sigma_r^{(\xi)})$. The following result provides an upper bound on the KL divergence between the distributions under parameters $\xi, \xi' \in \Xi^d$ differing in only a single coordinate.

**Claim 2.** *Let $\xi$ and $\xi'$ differ only in coordinate $i$ where $\xi_i = -\xi_i'$. Then*

$$D_{KL}(\mathbb{P}^{(\xi)} \| \mathbb{P}^{(\xi')}) \leq \frac{25}{16} \frac{h_r^2}{\rho_r^4}.$$

*Proof of Claim 2.* Observe that

$$D_{\mathrm{KL}}(\mathbb{P}^{(\xi)} \| \mathbb{P}^{(\xi')}) = \frac{1}{2} \left[ \log \frac{\det\left(\Sigma_r - h_r u_i u_i^\top\right)}{\det\left(\Sigma_r + h_r u_i u_i^\top\right)} - d + \mathrm{Tr}\left(\left(\Sigma_r - h_r u_i u_i^\top\right)^{-1}\right)\left(\Sigma_r + h_r u_i u_i^\top\right)\right]$$

Define $\alpha = h_r u_i^\top \Sigma_r^{-1} u_i$. By the matrix determinant lemma,

$$\det\left(\Sigma_r - h_r u_i u_i^\top\right) = (1 - \alpha)\det(\Sigma_r).$$

By Sherman Morrison, $\left(\Sigma_r - h_r u_i u_i^\top\right)^{-1} = \Sigma_r^{-1} \frac{h_r}{1-\alpha}(\Sigma_r^{-1} u_i u_i^\top \Sigma_r^{-1})$. Substituting and simplifying, we reach

$$D_{\mathrm{KL}}(\mathbb{P}^{(\xi)} \| \mathbb{P}^{(\xi')}) = \frac{1}{2} \left[ \log \frac{(1 - \alpha)}{(1 + \alpha)} + \frac{2\alpha}{1 - \alpha} \right].$$

Observe $\alpha \geq 0$ since $\Sigma_r$ is PSD. By (57), $\Sigma_r^{-1} \preceq \frac{1}{0.9\rho_r^2} I_d$ thus $\alpha \leq \frac{h_r}{0.9\rho_r^2}$. By (56), it follows that $0 \leq \alpha \leq \frac{1}{9d} \leq \frac{1}{9}$ for $d \geq 1$. Define $\gamma := \frac{2\alpha}{1-\alpha}$ where $\gamma \in [0, 1/4]$. Using the fact that, for $\gamma \in [0, 1/4]$, $-\log(1 + \lambda) + \lambda \leq \frac{\lambda^2}{2}$ and the fact that, for $\alpha \in (0, 1/9)$, $\frac{\alpha^2}{(1-\alpha)^2} \leq \frac{81}{64}\alpha^2$,

$$D_{\mathrm{KL}}(\mathbb{P}^{(\xi)} \| \mathbb{P}^{(\xi')}) = \frac{1}{2} \left[ -\log(1 + \lambda) + \lambda \right] \leq \frac{\lambda^2}{4} = \frac{\alpha^2}{(1 - \alpha)^2} \leq \frac{81}{64}\alpha^2 \leq \frac{81}{64} h_r^2 \left\|\Sigma_r^{-1}\right\|_2^2 \leq \frac{25}{16} \frac{h_r^2}{\rho_r^4}$$

where the last inequality follows by (57). $\qquad\square$

By Claim 1, over $n_r$ independent samples from group $r$,

$$D_{\mathrm{KL}}(\mathbb{P}^{(\xi)\otimes n_r} \| \mathbb{P}^{(\xi')\otimes n_r}) \leq \frac{25}{16} \frac{h_r^2 n_r}{\rho_r^4}.$$

By (56) and an application of Pinsker's inequality, we reach the bound

$$\left\|\mathbb{P}^{(\xi)\otimes n_r}, \mathbb{P}^{(\xi')\otimes n_r}\right\|_{\mathrm{TV}} \leq \frac{1}{2}. \tag{58}$$

**Parameter separation:**  Define $A^{(\xi)} := \left( \lambda \Sigma_r^{(\xi)} + (1-\lambda) \Sigma_b \right)$. The target parameter under $\xi$ can be expressed as

$$\beta_\lambda^{(\xi)} = \left( A^{(\xi)} \right)^{-1} \left( \lambda \Sigma_r^{(\xi)} \beta_r + (1-\lambda) \Sigma_b \beta_b \right) = \beta_r + \left( A^{(\xi)} \right)^{-1} (1-\lambda) \Sigma_b (\beta_b - \beta_r).$$

**Claim 3.** *Let $\xi$ and $\xi'$ differ only in coordinate $i$ where $\xi_i = -\xi_i'$. Then,*

$$\left| \left\langle e_i, \beta_\lambda^{(\xi)} - \beta_\lambda^{(\xi')} \right\rangle \right| \gtrsim \lambda(1-\lambda) h_r \rho_b^2 \rho_\lambda^{-4} \|\beta_r - \beta_b\|.$$

*Proof of Claim 3.*  By Sherman-Morrison,

$$A^{(\xi)-1} - A^{(\xi')-1} = \frac{2\lambda h_r}{1 + 2\lambda h_r u_i^\top A^{(\xi)-1} u_i} A^{(\xi)-1} u_i u_i^\top A^{(\xi)-1}.$$

Hence, we can express

$$\beta_\lambda^{(\xi)} - \beta_\lambda^{(\xi')} = \left( A^{(\xi)-1} - A^{(\xi')-1} \right) (1-\lambda) \Sigma_b (\beta_b - \beta_r) \Sigma_b (\beta_b - \beta_r).$$

Then, substitution of $v$ defined in (55a),

$$\langle e_i, \beta_\lambda^{(\xi)} - \beta_\lambda^{(\xi')} \rangle = 2\lambda(1-\lambda) h_r \|\Sigma_b\| \|\beta_r - \beta_b\| \frac{e_i^\top A^{(\xi)-1} u_i u_i^\top A^{(\xi)-1} v}{1 + 2\lambda h_r u_i^\top A^{(\xi)-1} u_i}.$$

By the Loewner order relationship (57),

$$\left\| A^{(\xi)} \right\| \geq 0.9\lambda\rho_r^2 + (1-\lambda)\rho_b^2 \geq 0.9\lambda\rho_r^2$$

The denominator is bounded, therefore, by:

$$\left| 1 + 2\lambda h_r u_i^\top A^{(\xi)-1} u_i \right| \leq \left| 1 + 2\lambda h_r \left\| A^{(\xi)-1} \right\| \right| \leq \left| 1 + \frac{2\lambda h_r}{(0.9\lambda\rho_r^2)} \right| \leq \frac{11}{9}$$

where the last inequality follows by assuming $h_r \leq \frac{\rho_r^2}{10d}$ and $d \geq 1$.

Next, we lower bound the numerator. Fix a coordinate $i \in [d]$, and observe that we can write the decomposition

$$A^{(\xi)-1} e_i = c_1 e_i + c_2 w_i$$

where $w_i \in \mathbb{R}^d$ is orthogonal to $e_i$ and satisfies $\|w_i\| = 1$. By definition, $c_1 = e_i^\top A^{(\xi)-1} e_i$, and, using (57),

$$\frac{1}{1.1}\rho_\lambda^{-2} \leq \left( 1.1\lambda\rho_r^2 + (1-\lambda)\rho_b^2 \right)^{-1} \leq c_1 \leq \left( 0.9\lambda\rho_r^2 + (1-\lambda)\rho_b^2 \right)^{-1} \leq \frac{1}{0.9}\rho_\lambda^{-2}. \tag{59}$$

Moreover, by construction and again by (57),

$$c_1^2 + c_2^2 = \left\| A^{(\xi)-1} e_i \right\|^2 \leq \left\| A^{(\xi)-1} \right\|^2 \leq \left( 0.9\lambda\rho_r^2 + (1-\lambda)\rho_b^2 \right)^{-2} \leq \frac{1}{0.9^2}\rho_\lambda^{-4}. \tag{60}$$

Combining (59) and (60), we have

$$c_2 \leq \left( 0.9^{-2} - 1.1^{-2} \right)^{0.5} \rho_\lambda^{-2}. \tag{61}$$

Next, noting that the vectors $u_i$ and $v$ satisfy $|u_i^\top e_i|, |u_i^\top v| \geq \frac{1}{\sqrt{2}}$, we have

$$|e_i^\top A^{(\xi)-1} u_i| = |c_1 e_i^\top u_i + c_2 w_i^\top u_i| \geq \frac{c_1}{\sqrt{2}} - |c_2|, \quad |u_i^\top A^{(\xi)-1} v| \geq \frac{c_1}{\sqrt{2}} - |c_2|.$$

Combining (59) and (61), we see that

$$\frac{c_1}{\sqrt{2}} - |c_2| \geq \frac{1}{300}\rho_\lambda^{-2}.$$

All together, this yields the parameter separation lower bound:

$$\langle e_i, \beta_\lambda^{(\xi)} - \beta_\lambda^{(\xi')} \rangle^2 \geq \left( \frac{18}{11 \cdot 300^2} \right)^2 \lambda^2 (1-\lambda)^2 h_r^2 \rho_b^4 \rho_\lambda^{-8} \|\beta_r - \beta_b\|^2.$$

$\square$

**Assouad's Lemma:** Let $\widehat{\beta}$ be any estimator. Assouad's lemma (see Lemma 6) applied to the $d$-dimensional hypercube $\Xi_d$, given the results Claim 3 and (58), yields

$$\sup_{\xi \in \Xi^d} \mathbb{E}_\xi \left[ \left\| \widehat{\beta} - \beta \right\|_2^2 \right] \geq \left( \frac{18}{11 \cdot 300^2} \right)^2 \frac{2}{25} \lambda^2 (1-\lambda)^2 \frac{\rho_r^4 \cdot \rho_b^4 \cdot d}{\rho_\lambda^8 \cdot n_r} \left\| \beta_r - \beta_b \right\|^2 . \tag{62}$$

**Symmetric perturbation for group $b$:** Repeating the procedure so far but instead perturbing the covariance structure of group $b$, i.e., taking, for a Rademacher vector $\zeta \in \{-1, 1\}^d$, set

$$\Sigma_r := \rho_r^2 I_d, \quad \Sigma_b^{(\zeta)} := \rho_b^2 I_d + h_b \sum_{i=1}^d \xi_i u_i u_i^\top, \quad h_b = \frac{2\rho_b^2}{5\sqrt{n_b}}.$$

This construction yields the analogous lower bound

$$\sup_{\zeta \in \Xi^d} \mathbb{E}_\xi \left[ \left\| \widehat{\beta} - \beta \right\|_2^2 \right] \geq \left( \frac{18}{11 \cdot 300^2} \right)^2 \frac{2}{25} \lambda^2 (1-\lambda)^2 \frac{\rho_r^4 \cdot \rho_b^4 \cdot d}{\rho_\lambda^8 \cdot n_b} \left\| \beta_r - \beta_b \right\|^2 . \tag{63}$$

Since we may perturb either group, we may take the maximal lower bound and use the fact that $\max(a, b) \geq \frac{1}{2}(a + b)$ to obtain the bound:

$$\sup_{\xi, \zeta \in \Xi^d} \left[ \left\| \widehat{\beta} - \beta \right\|_2^2 \right] \geq C \lambda^2 (1-\lambda)^2 \frac{\rho_r^4 \cdot \rho_b^4 \cdot d}{\rho_\lambda^8} \left( \frac{1}{n_r} + \frac{1}{n_b} \right) \left\| \beta_r - \beta_b \right\|^2$$

where $C = \left( \frac{18}{11 \cdot 300^2} \right)^2 \frac{1}{25}$.

Finally, using that $\left\| \widehat{\beta}_\lambda - \beta_\lambda \right\|_{\Sigma_g} \geq \rho_g^2 \left\| \widehat{\beta}_\lambda - \beta_\lambda \right\|_2^2$, we reach the bound

$$\mathbb{E} \left[ \left\| \widehat{\beta}_\lambda - \beta_\lambda \right\|_{\Sigma_g}^2 \right] \geq \lambda^2 (1-\lambda)^2 \frac{\rho_g^2 \cdot \rho_r^4 \cdot \rho_b^4 \cdot d}{\rho_\lambda^8} \left( \frac{1}{n_r} + \frac{1}{n_b} \right) \left\| \beta_r - \beta_b \right\|^2 . \quad \blacksquare$$

### C.3.4 Proof of Proposition 3

First, note that, using (50) from the proof of Theorem 3, we have

$$\left| \mathbb{E} \left[ (\widehat{\beta}_\lambda - \beta_\lambda)^\top \Sigma_r (\beta_\lambda - \beta_r) \right] \right| \lesssim \frac{(1-\lambda)\rho_r^2 \rho_b^2}{\lambda \rho_r^2 + (1-\lambda)\rho_b^2} \left\| \beta_r - \beta_b \right\| \left\| \mathbb{E} \left[ \widehat{\beta}_\lambda - \beta_\lambda \right] \right\| . \tag{64}$$

Next, we recall from the proof of Lemma 2 that

$$\mathbb{E} \left[ \widehat{\beta}_\lambda - \beta_\lambda \right] = \mathbb{E} \left[ \widehat{\Sigma}_\lambda^{-1} \left( \lambda \widehat{\Sigma}_r \beta_r + (1-\lambda) \widehat{\Sigma}_b \beta_b \right) - \beta_\lambda \right] , \tag{65}$$

which, as described in the proof of Proposition 2, can be further cast as

$$\mathbb{E} \left[ \widehat{\beta}_\lambda - \beta_\lambda \right] = \mathbb{E} \left[ \widehat{\Sigma}_\lambda^{-1} \left( \lambda \left( \widehat{\Sigma}_r - \Sigma_r \right) (\beta_r - \beta_\lambda) + (1-\lambda) \left( \widehat{\Sigma}_b - \Sigma_b \right) (\beta_b - \beta_\lambda) \right) \right] \tag{66}$$

Woodbury matrix identity, we can write

$$\widehat{\Sigma}_\lambda^{-1} = \Sigma_\lambda^{-1} - \Sigma_\lambda^{-1} (\widehat{\Sigma}_\lambda - \Sigma_\lambda) \widehat{\Sigma}_\lambda^{-1}. \tag{67}$$

By substituting the above identity into (66), and using the fact that $\mathbb{E}[\widehat{\Sigma}_g - \Sigma_g] = 0$ for both groups, we obtain

$$\mathbb{E} \left[ \widehat{\beta}_\lambda - \beta_\lambda \right] = -\mathbb{E} \left[ \Sigma_\lambda^{-1} (\widehat{\Sigma}_\lambda - \Sigma_\lambda) \widehat{\Sigma}_\lambda^{-1} \left( \lambda \left( \widehat{\Sigma}_r - \Sigma_r \right) (\beta_r - \beta_\lambda) + (1-\lambda) \left( \widehat{\Sigma}_b - \Sigma_b \right) (\beta_b - \beta_\lambda) \right) \right] . \tag{68}$$

Next, we bound the two terms on the right hand side separately. First, notice that

$$\left\| \mathbb{E} \left[ \Sigma_\lambda^{-1} (\widehat{\Sigma}_\lambda - \Sigma_\lambda) \widehat{\Sigma}_\lambda^{-1} \left( \widehat{\Sigma}_r - \Sigma_r \right) (\beta_r - \beta_\lambda) \right] \right\|$$

$$\lesssim \frac{(1-\lambda)\rho_b^2}{(\lambda \rho_r^2 + (1-\lambda)\rho_b^2)^2} \left\| \beta_r - \beta_b \right\| \left\| \mathbb{E} \left[ (\widehat{\Sigma}_\lambda - \Sigma_\lambda) \widehat{\Sigma}_\lambda^{-1} \left( \widehat{\Sigma}_r - \Sigma_r \right) \right] \right\|$$

$$\lesssim \frac{(1-\lambda)\rho_b^2}{(\lambda \rho_r^2 + (1-\lambda)\rho_b^2)^2} \left\| \beta_r - \beta_b \right\| \sqrt{\mathbb{E} \left[ \left\| \widehat{\Sigma}_\lambda^{-1} \right\|^2 \right]} \left( \mathbb{E} \left[ \left\| \Sigma_r - \widehat{\Sigma}_r \right\|^4 \right] \right)^{1/4} \left( \mathbb{E} \left[ \left\| \Sigma_\lambda - \widehat{\Sigma}_\lambda \right\|^4 \right] \right)^{1/4}$$

$$\lesssim \frac{(1-\lambda)\rho_b^2}{(\lambda \rho_r^2 + (1-\lambda)\rho_b^2)^2} \left\| \beta_r - \beta_b \right\| \frac{K_r^2 \rho_r^2 \sqrt{d}/\sqrt{n_r}}{\lambda \rho_r^2/C_r' + (1-\lambda)\rho_b^2/C_b'} \left( \lambda K_r^2 \rho_r^2 \frac{\sqrt{d}}{\sqrt{n_r}} + (1-\lambda) K_b^2 \rho_b^2 \frac{\sqrt{d}}{\sqrt{n_b}} \right), \tag{69}$$

where the last inequality follows from Lemma 3 and Lemma 4. Similarly, we can show

$$\left\| \mathbb{E}\left[ \Sigma_\lambda^{-1}(\widehat{\Sigma}_\lambda - \Sigma_\lambda)\widehat{\Sigma}_\lambda^{-1}\left(\widehat{\Sigma}_b - \Sigma_b\right)(\beta_b - \beta_\lambda)\right]\right\|$$

$$\lesssim \frac{\lambda\rho_r^2}{(\lambda\rho_r^2 + (1-\lambda)\rho_b^2)^2} \left\| \beta_r - \beta_b\right\| \frac{K_b^2\rho_b^2\sqrt{d}/\sqrt{n_b}}{\lambda\rho_r^2/C_r' + (1-\lambda)\rho_b^2/C_b'}\left(\lambda K_r^2\rho_r^2\frac{\sqrt{d}}{\sqrt{n_r}} + (1-\lambda)K_b^2\rho_b^2\frac{\sqrt{d}}{\sqrt{n_b}}\right).$$

$$(70)$$

Plugging (69) and (70) into (68), and then substituting the whole term into (64) completes the proof.
∎.

