# OpenReview forum: "The Statistical Fairness-Accuracy Frontier"
_NeurIPS.cc/2025/Workshop/Reliable_ML — NeurIPS 2025 - Reliable ML Workshop_

### Official Review · Reviewer_YvyC · 2025-09-14
**This is a valuable theoretical contribution, but empirical support and broader accessibility would strengthen it.**

**Rating:** 7
**Confidence:** 3

**Review:**

Summary:
This paper studies the fairness-accuracy(FA) frontier in machine learning, focusing on the finite-sample regime. Prior work (e.g., Liang et al. 2021) characterized the FA frontier assuming full knowledge of population distributions, this paper analyzes how sampling variability and heterogeneous group distribution distort the empirical frontier through the framework of linear regression.

Strengths:
- The theory is rigorous and solid. Both upper and lower bounds for bias and variance terms are provided.
- The FA frontier is a clean conceptual framework; the author extend it carefully to finite-sample considerations.
- Some results are translated into actionable guidelines for sample allocation (e.g., proportional to $\lambda \rho r/((1-\lambda)\rho b)$).

Weakness:
- The paper focuses exclusively on linear regression. While this makes analysis tractable, it limits direct applicability to classification and modern ML models.

Suggestions:
- Discuss extensions beyond linear regression (e.g., classification, generalized linear models).
- Expand limitations and broader impacts: how might these findings inform regulatory or practitioner decisions?

---

### Official Review · Reviewer_Uznh · 2025-09-21
**Linear regression on the fairness accuracy frontier**

**Rating:** 8
**Confidence:** 3

**Review:**

## Summary

This paper considers the set of models on the fairness accuracy pareto frontier. This frontier consists of all models where neither fairness nor accuracy can improved without harming the other. The paper considers a specific setting with two groups and linear regression. We seek to minimize the excess fair risk (loss), that is, the difference between the empirical fairr loss and the unobtainable population loss.

First, the paper considers the setting where the covariance matrices are known for each group in {r, b}. Here, in Theorem 1, the paper proposes a minimax optimal estimator $\hat{\beta}_\lambda$ which is simply the empirical version of the weighted average population risk between the two groups (where the weight is $\lambda$).

The next setting is one with unknown covariances. Here, the picture is more difficult. First, proposition 2 is introduced which bounds the distance between $\hat{\beta}\lambda$ and its population counterpart $\beta_\lambda$ via a bias and variance term. The variance term is similar to the known covariance matrix case. The bias term, however, depends on the \emph{difference} between $\beta_r$ and $\beta_b$, which dissapears to zero at equality (similar to the known covariance case). Finally, Theorem 2 and 3 place more quantitative bounds on the covariance and bias in terms of known quantities.

A key aspect throughout is: how much of each group should we sample? In the known covariance case, an optimal sampling scheme is one which chooses $n_r / n_b$; Alternatively, we should advocate for a balanced design with unknown covariances.

## Strengths

Interesting analysis of the fairness accuracy frontier. Cool results on the intuition of when we should sample from groups in a balanced way vs. not. Interesting that this depends on whether the covariance matrices are known!

## Weaknesses

I think that some notation is not defined in the main paper, I was unable to find definitions of $n_r$ (I am assuming this is number of samples from the red group) or $\rho_r$ (fraction of the red group).

## Suggestions

I am sure that future reviewers will ask for simple experiments, I don't think they are necessary because the theory is already interesting, but it it something to keep in mind.

Overall, a strong technical paper worth reading and understanding further.